



# Retrieving tropospheric NO₂ vertical column densities around the city of Beijing and estimating NOₓ emissions based on car MAX-DOAS measurements

Xinghong Cheng[1], Jianzhong Ma[1], Junli Jin[2], Junrang Guo[1], Yuelin Liu[3], Jida Peng[4],
Xiaodan Ma[5], Minglong Qian[6], Qiang Xia[6], Peng Yan[2]

[1]State Key Lab of Severe Weather & Key Laboratory for Atmospheric Chemistry, Chinese Academy of Meteorological Sciences, Beijing 100081, China

[2]CMA Meteorological Observation Centre, Beijing 100081, China

[3]College of Architecture and Environment, Sichuan University, Chengdu 610065, China

[4]Meteorological Institute of Fujian, Fuzhou 350001, China

[5]Nanjing University of Information Science and Technology, Nanjing 210044, China

[6]China National Huayun Technology Development Corporation, Beijing 100081, China

*Correspondence to:* Xinghong Cheng (cxingh@cma.gov.cn) and Jianzhong Ma (majz@cma.gov.cn)

**Abstract.** We carried out 19 city-circle-around car MAX-DOAS experiments on the 6th Ring Road of Beijing in January, September, and October 2014. The tropospheric vertical column densities (VCDs) of NO₂ were retrieved from measured spectra obtained by the multi-axis differential optical absorption spectroscopy (MAX-DOAS) technique and used to estimate the emissions of NOₓ ($\equiv$ NO + NO₂) from urban Beijing during the experimental periods. The offline LAPS-WRF-CMAQ model system was used to simulate the wind fields by assimilation of observational data and calculate the NO₂-to-NOₓ concentration ratios, both of which are also needed for the estimation of NOₓ emissions. The NOₓ emissions in urban Beijing for the different seasons derived from the car MAX-DOAS measurements in this study were compared to the multi-resolution emission inventory in China for 2012 (MEIC 2012). Our car MAX-DOAS measurement results showed higher NO₂ VCD in January than in the other two months and typically larger NO₂ VCD at the southern parts of the 6th Ring Road than at the northern parts. The wind field had obvious impacts on the spatial distribution of NO₂ VCD, with the mean NO₂ VCD along the 6th Ring Road typically being higher under the south wind than under the north wind. In addition to the seasonal difference, the journey-to-journey variations of estimated NOₓ emissions rates (E$_{NOX}$) were large even within the same month, mainly due to uncertainties in the calculations of wind speed, the ratio of NO₂ and NOₓ concentration, and the decay rate of NOₓ from the emission sources to the measured positions under different meteorological conditions. The ranges of E$_{NOX}$ during the heating



and non-heating periods were $28.7 \times 10^{25}$ to $60.0 \times 10^{25}$ molecules $s^{-1}$ and $7.7 \times 10^{25}$ to $24.8 \times 10^{25}$ molecules $s^{-1}$, respectively. The average $E_{NOX}$ values in the heating and non-heating periods were $43.0 \times 10^{25}$ molecules $s^{-1}$ and $13.9 \times 10^{25}$ molecules $s^{-1}$, respectively. The uncertainty range of $E_{NOX}$ was 16.4–33.2%. The monthly emission rates from MEIC 2012 are found to be lower than the estimated $E_{NOX}$,

particularly in January. Our results provide important information and datasets for the validation of satellite products and also show how car MAX-DOAS measurements can be used effectively for dynamic monitoring and updating of the $NO_x$ emissions from megacities such as Beijing.

## 1.      Introduction

Over the past decade, serious haze weather has occurred frequently in autumn and winter in Beijing due to massive anthropogenic emissions from the consumption of fossil fuels and other sources (He et al., 2013; Zhang et al., 2013). High concentrations of aerosol particulate matter with dynamic diameter smaller than 2.5 μm ($PM_{2.5}$) threaten public health (Cao et al., 2014), disturb traffic operation by affecting visibility, and result in changes to the weather and climate because of

scattering and absorption of solar radiation (Liao et al., 2015; Cheng et al., 2017). Measurements have shown that organic matter (OM), sulfate, nitrate, and ammonium made up more than 78% of the $PM_{2.5}$ in January 2013 in Beijing (Huang et al., 2014). Fractions of nitrate in $PM_{2.5}$ have obviously increased recently with the control of industry and coal in the Beijing-Tianjin-Hebei region, which has reduced $SO_2$ emissions and the ratio of sulfate in $PM_{2.5}$. Recent research (Tan et al., 2018) based on the aerosol

observations at the campus of Peking University in 2014 revealed that aerosol pollution is nitrate-driven in spring and early fall and OM-driven in late fall and winter. The researchers suggested that nitrate formation was more significant than sulfate formation during severe pollution episodes in Beijing. Therefore, studies on the spatiotemporal variation of gaseous precursors of nitrate are very important for understanding the aerosol formation and its influencing factors. NO and $NO_2$ (together denoted as $NO_X$)

form primarily in combustion processes, and the conversion between NO and $NO_2$ in the atmosphere is very rapid.

     Emission inventories are usually developed by the so-called bottom-up approach, which is based on combinations of activity statistics (such as energy consumption and industrial production) and source- or region-specific emission factors (Hao et al., 2002;Zhang et al., 2007;Zhao et al., 2012;Streets et al., 2013).

However, there are high uncertainties in bottom-up emissions inventories associated with the statistics,



emissions factors, temporal allocation profiles, and grid allocation factors (Ma and Van Aardenne, 2004; Zhao et al., 2012). Moreover, estimating "current" emissions by the bottom-up methodology is fundamentally difficult because publication of basic statistics is generally a couple of years behind. The top-down constraint is a useful supplement to bottom-up estimates, which are subject to uncertainties in

emissions factors and emissions activities (Streets et al., 2013). Inverse modeling, in which emissions are optimized to reduce the differences between simulated and observed data, is a powerful method that solves the problems of the bottom-up approach. Recently, its application to the estimation of $NO_X$ emissions has been widely reported. $NO_X$ emission rates are derived by constraining satellite observations using the relationship between model-simulated $NO_2$ vertical column density (VCD) and

primary $NO_X$ emission estimates from the bottom-up approach (Martin, 2002; Jaegle' et al., 2005; Konovalov et al., 2006; Wang et al., 2007; Lin et al., 2012; Zyrichidou et al., 2015).Nevertheless, errors and uncertainties still exist in the retrieval of satellite data, and these lead to a large decrease in precision, particularly in highly polluted regions such as Beijing and its surroundings (Ma et al., 2013a;Jin et al., 2016). Uncertainties can arise from noise, surface albedo, cloud blocks, profile shape, interference from

ozone absorption, correlations with other retrieved parameters, fitting wavelength window, and so forth. Air mass factor (AMF) errors can produce additional errors during the conversion process from the slant to vertical columns. Therefore, comprehensive ground-based measurements of the tropospheric columns and vertical profiles of $NO_2$ are quite important and necessary to evaluate and validate satellite retrieval products.

MAX-DOAS (Multi-Axis Differential Optical Absorption Spectroscopy) is a new ground-based remote sensing technique developed during the last decade. It makes use of the scattered sunlight measured from horizontal through zenith direction to retrieve the VCD and profiles of trace gases and aerosols with relatively high sensitivity in the lower atmosphere (Hönninger et al., 2004;Wagner et al., 2004;Platt and Stutz, 2008). MAX-DOAS has been extensively used to derive tropospheric column

information of $NO_2$ and some other pollutants in various regions(Wittrock et al., 2004; Brinksma, et al., 2008; Irie et al., 2008;Vlemmix et al., 2010;Li et al., 2013;Hendrick et al., 2014). Mobile- (or car-) MAX-DOAS measurements have been used to quantify $NO_X$ emissions from cities and regions such as Beijing (Johansson et al., 2008), Mexico (Johansson et al., 2009), Mannheim and Ludwigshafen (Ibrahim et al., 2010), Deli (Shaiganfar et al., 2011), Shanghai (Wang et al., 2012), North China (Wu et al., 2018).

Compared to ground-based observations at a fixed site, car-MAX-DOAS measurements can provide





information on the spatial distribution of pollutants, which is important for explaining the urban/regional

representativeness of satellite observations over megacities such as Beijing. Moreover, due to the rapid

expansion of urban area and increasing energy consumption, both locations and strength of emission

sources in Beijing may have changed significantly. Therefore, intensive Car-MAX-DOAS measurement

campaigns are still needed to estimate the emissions of $NO_X$ in Beijing. Mean wind speed and wind

direction along the ring road during the sampling periods are usually used to estimate $NO_X$ emissions in

the previous studies. Since wind field changes rapidly due to local circulation and then results in

uncertainties in quantification of $NO_X$ emissions (Johansson et al., 2008; Shaiganfar et al., 2011, 2017;

Davis, et al., 2019), refined and accurate simulations of wind fields are needed for the accurate emission

estimate.

In this study, we estimated the total $NO_X$ emissions from urban Beijing based on the VCD of $NO_2$

obtained from intensive car MAX-DOAS measurements on the 6th Ring Road of Beijing in January,

September, and October of 2014. The offline LAPS-WRF-CMAQ model system with data assimilation

method was used to derive wind speed, wind direction, and $NO_2/NO_X$ concentration ratios, which are

needed to estimate total urban $NO_X$ emissions based on car MAX-DOAS measurements. This paper is

organized as follows: Section 2 describes the intensive car MAX-DOAS experiment and the retrieval

method for deriving tropospheric $NO_2$ VCD, the model system used to simulate wind field and the ratios

of $NO_2$ and $NO_X$, and the method used to quantify total $NO_X$ emissions. Section 3 presents the results of

the $NO_2$ VCD and the estimated $NO_X$ emissions as well as their uncertainties due to simulated errors in

the wind field. Conclusions are provided in Section 4.

## 2.       Theory, experimental, and method
### 2.1       Formula to estimate urban NOx emissions

The complete $NO_2$ flux $F_{NO_2}$ across the urban Beijing area encircled by the driving route $S$ is estimated

according to the method of Ibrahim et al. (2010).

$$F_{NO_2} = \oint_s VCD_{NO_2}(s) \cdot \overline{w} \cdot \overline{n} \cdot ds \qquad (1)$$

Here $VCD_{NO_2}(s)$ is the $NO_2$ VCD at the sampling position within the driving route; $\overline{n}$ indicates the

normal vector parallel to the Earth's surface and orthogonal to the driving direction at the position of the

driving route; $\overline{w}$ is the average wind vector within the $NO_2$ layer, which is denoted by wind speed at the

height of 10 m. We carried out car MAX-DOAS measurements along closed driving routes around large





emissions sources, i.e., the 6th Ring Road of Beijing (Fig. 1).

According to the calculation method of Ibrahim et al. (2010), the complete $NO_X$ emissions from the encircled areas are determined considering the partitioning between NO and $NO_2$ ($c_L$) and the finite lifetime of $NO_X$ ($c_\tau$).


$$E_{NO_x} = c_L \cdot c_\tau \cdot F_{NO_2} \tag{2}$$

$$c_L = \frac{c_{NO_x}}{c_{NO_2}} \tag{3}$$

Here $c_L$ is simply the ratio of $NO_X$ ($C_{NO_x}$) and $NO_2$ ($C_{NO_2}$) bulk concentration in the polluted layer which are simulated by the CMAQ model in this study. It is a function of the Leighton ratio (L=[NO]/[$NO_2$]), $c_L$=1+L. $c_\tau$ describes the decay of $NO_X$ from the emission sources to measured positions. $c_\tau$ can be


estimated from the $NO_X$ lifetime $\tau$, which is the reciprocal of the product of reaction rate coefficient $k$, OH concentration ($C_{OH}$) and air density (M)(Ma et al., 2013), and transport time t, which is the distance between emission source and sampling point r divided by the wind speed w.

$$c_\tau = e^{\frac{t}{\tau}} = e^{\frac{r/w}{\tau}} \tag{4}$$

$$\tau = \frac{1}{k * C_{OH} * M} \tag{5}$$


We averaged our model simulated quantities over the urban area for the $NO_x$ lifetime $\tau$, used the simulated wind speed at sampling position as w, and computed the distance between the sampling position and the center of the city of Beijing for r.

### 2.2 Car MAX-DOAS measurements

#### 2.2.1 Instrument and experiment


We measured and retrieved tropospheric $NO_2$ VCD along the sixth ring road of Beijing (hereafter referred to as 6th Ring Rd) in January, September, and October of 2014 using a Mini MAX-DOAS instrument settled on the vehicle.

The instrument, manufactured at Hoffmann Messtechnik GmbH, Germany, is a fully automated, light-weighted spectrometer designed for the spectral analysis of scattered sunlight by the MAX-DOAS


technique (Hönninger et al., 2004). The same type of instrument was used in previous studies, including long-term site measurements in Beijing (Ma et al., 2013a)and a car MAX-DOAS observational journey in Europe (Wagner et al., 2010a). For this study, the instrument was mounted on the roof a car. Inside the car, two 12V DC batteries alternatively supplied electronic power for the running of instruments and





a laptop computer, with a script run by the DOASIS software (Kraus, 2001b) to control the measurement

process and the recording of spectra. The temperature of the spectrograph was set to be maintained at –5°C in January and at 0°C in September and October, well below the ambient temperatures during the experimental days of the study. The signal spectra of dark current and electronic offset were measured each day before and after the field experiment on the road, with 10000 msec and 1 scan for dark current measurements and 3 msec and 1000 scans for electronic offset measurements. Measurements were made

alternatively at 30° and 90° elevation angles, with an integration time of about 1 min for each elevation angle.

The instrument onboard the car was operated to measure scattered sunlight from the driving forward direction. There were no high buildings on both sides of the 6th Ring Rd., and the measurements were made at a wide-field view. The driving speed was typically controlled at 80–90 km h$^{-1}$, and it generally

took about 2.0–2.5 h to complete one circle (about 187 km) around the 6th Ring Rd. Figure 1 shows the driving route of the car MAX-DOAS experiment on a map of Beijing. For this study, the field experiments were carried out on 14 selected days, with one or two circle journeys each day. In total, there are 19 circling journeys available. The sampling periods in this experiment and the meteorological conditions are listed in Table 1. In most cases, the meteorological conditions changed slightly within one

circling journey period. The average wind speeds for experimental days in January, September, and October were 2.5, 2.5, and 2.4 m s$^{-1}$, the corresponding total cloud fractions were 4.9, 7.5, and 4.2, and the mean planetary boundary layer (PBL) heights were 192, 188, and 238 m, respectively. The dominant wind directions in the three months were much more variable, including north, south, and other directions.

### 2.2.2 Spectral retrieval

The retrieval of NO$_2$ slant column densities (SCDs) is based on the DOAS method (Platt, 1994). The WinDOAS software (Fayt and Van Roozendael, 2011) was adopted to analyze the spectra in the 400-431 nm range on a daily basis. The Fraunhofer reference spectrum (FRS) was selected among the measured spectra at the 90° elevation angle each day by two steps: first, a spectrum measured around noon was chosen; second, the spectrum corresponding to the minimum NO$_2$ SCD derived in the

preliminary analysis using the FRS from the first step was finally selected. The cross sections of NO$_2$ at 294 K (Vandaele et al., 1998), O$_3$ at 221 K (Burrows et al., 1999), and the Oxygen dimmer O$_4$ at 298 K (Greenblatt et al., 1990), as well as a FRS, a Ring spectrum calculated from the FRS by DOASIS (Kraus,





2001a) and a polynomial of third order were included in the spectral fitting process. Figure 2 shows an

example of our spectral analysis for a measurement on 18 January 2014, 11:39:38 BJT. As shown in the

figure, the atmospheric $NO_2$ absorption structure can be clearly extracted from the measured spectra.

### 2.2.3 Derivation of tropospheric NO₂ VCD

The trace gas VCD in the troposphere can be calculated using its SCD divided by the air mass factor

(AMF) at an elevation angle, $\alpha$:

$$VCD_{trop} = \frac{SCD_{trop}(\alpha)}{AMF_{trop}(\alpha)} \qquad (6)$$

For the site MAX-DOAS measurements, a FRS from the same elevation sequence was used in most

cases, and the stratospheric absorption can be assumed to be the same during one elevation sequence.

Therefore, the $VCD_{trop}$ can be calculated by extending Eq. 1 to Eq. 2 using the so-called differential

tropospheric slant column density ($DSCD_{trop}(\alpha) = SCD_{trop}(\alpha)–SCD_{trop}(90°)$) divided by the differential

air mass factor ($DAMF_{trop}(\alpha) = AMF_{trop}(\alpha)–AMF_{trop}(90°)$):

$$VCD_{trop} = \frac{DSCD_{trop}(\alpha)}{DAMF_{trop}(\alpha)} = \frac{DSCD_{meas}(\alpha)}{DAMF_{trop}(\alpha)} \qquad (7)$$

with $DSCD_{meas}(\alpha) = SCD_{meas}(\alpha) – SCD_{ref}$ (Wagner et al., 2010b;Ma et al., 2013a).

For the car MAX-DOAS measurements, the trace gas concentrations can change significantly during

one measurement sequence and thus the dependence of retrieved trace gas DSCDs on the elevation angle

may not be so regular as for the site measurements. Therefore, it would be a better choice to use a single

FRS for the analysis of all the spectra measured along the driving route (Wagner et al., 2010b). According

to Wagner et al. (2010b), Eq.1 can be further extended to

$$VCD_{trop} = \frac{DSCD_{meas}(\alpha) - DSCD_{offset}}{AMF_{trop}(\alpha)} \qquad (8)$$

where $DSCD_{offset}$ depends on the solar zenith angle (SZA) and thus local time, $t_i$. For each elevation

sequence i during the individual measurement day, $DSCD_{offset}$ is calculated from a single pair of

measurements with

$$DSCD_{offset}(t_i) = \frac{AMF_{trop}(90°) \cdot DSCD_{meas}(\alpha,t_i) - AMF_{trop}(\alpha) \cdot DSCD_{meas}(90°,t_i)}{AMF_{trop}(\alpha) - AMF_{trop}(90°)} \qquad (9)$$

The time series of the calculated $DSCD_{offset}(t_i)$ in this study could be fitted by a low-order polynomial,

e.g., $P(x) = a_0 + a_1 \cdot x + a_2 \cdot x^2$, as a function of time. The fitted polynomial then represents the best guess

for $DSCD_{offset}$ and can be used to calculate the $VCD_{trop}$ from Eq. 3. In this study, the AMF was calculated

by the geometry approximation, that is:





$$\text{AMF}_{\text{trop}}(\alpha) \approx \frac{1}{\sin(\alpha)} \qquad\qquad (10)$$

As an illustration, Figure 3 shows the changes of individual $NO_2$ $DSCD_{\text{meas}}$ and $DSCD_{\text{offset}}$ for 30° elevation angle of each sequence as a function of time on 18 January 2014. A second order polynomial fitted from individual $DSCD_{\text{offset}}$ data points as shown in Fig. 3 tends to converge against a much more stable average $DSCD_{\text{offset}}$ value.

### 2.2.4 Calculation of monthly average NO2 VCD

To investigate the differences in the spatial distribution of $NO_2$ VCD among the three months, we computed the monthly average $NO_2$ VCD for every sampling point along the 6th Ring Rd of Beijing in January, September, and October, 2014. Firstly, we used the locations of all sampling points on the morning of September 23as the reference point for the calculation of $NO_2$ VCD monthly average, with the most sampling sites (98 points) for all observation periodsThen, we calculated the monthly average value at each reference point using the data of the nearest sampling point. The distance from the nearest sampling point to a reference point was less than 1.5 km.

### 2.3  LAPS-WRF-CMAQ model simulation

#### 2.3.1 Model setup and data

To quantify the $NO_X$ emissions in Beijing more accurately, refined simulations of the wind field and $NO_2$ to NOx concentration ratio were needed. In this study, we utilized the offline LAPS-WRF-CMAQ model system with high spatiotemporal resolution and data assimilation technique to obtain the refined wind speed and wind direction and an accurate ratio of $NO_2$ and $NO_X$ concentration during the car MAX-DOAS experiments.  The aforementioned model system includes three components: the LAPS model (Albers et al., 1996), the WRF model (Michalakes et al., 2004), and the CMAQ model (Dennis et al., 1996). Simulation of wind speed and direction is improved by the LAPS-WRF model, which assimilates observed data at the surface and high layers using the one-dimensional and three-dimensional variational assimilation method (Albers et al., 1996). The CMAQ model is used to simulate temporal-spatial distribution of $NO_2$ and NO concentration. The Local Analysis and Prediction System (LAPS), developed by the NOAA Earth System Research Laboratory, is used in many numerical weather forecast centers around the world. It is a mesoscale meteorological data assimilation tool that employs a suite of observations to generate a realistic, spatially distributed, time-evolving, threedimensional representation


of atmospheric features and processes (McGinley et al., 1991). The three-dimensional realistic

meteorological analyses field can be used as the initial condition of the WRF model and improve the

simulation of wind field. WRF is a mesoscale numerical weather prediction system designed for both

atmospheric research and operational forecasting needs. CMAQ is an air-quality model developed by the

U.S. Environmental Protection Agency's Atmospheric Science Modeling Division. It consists of a suite

of computer programs for modeling air quality issues, including reactive gases such as $NO_2$, $NO$, $SO_2$,

$O_3$, and others, particulate matter (PM), air toxics, acid deposition, and visibility degradation.

This study focused on Beijing at a horizontal resolution of 4 km × 4 km with 31 vertical layers of

varying thickness (between the surface and 50 hPa) using a triple-nested simulation technique. The

horizontal resolutions of the three sets of grids were 36 km, 12 km, and 4 km, respectively (Fig. S1a),

and the output temporal interval was 1 h. The LAPS-WRF simulations were driven by FNL/NCEP

analysis data every 6 h during the car MAX-DOAS experiments, with a spatial resolution of 1° × 1°. In

addition, to improve the simulation of wind field and $NO_2$ and $NO$ concentrations, many meteorological

data of the same periods, such as wind speed, wind direction, air temperature, and relative humidity,

observed at 2400 surface weather stations and by 120 radiosonde stations were assimilated into the initial

field of the WRF model using the one-dimensional and three-dimensional variational assimilation

method in the LAPS model. The CMAQ model uses the MEIC 2012 with 0.25° × 0.25° resolution (Zhang

et al., 2009;Li et al., 2017).. Hourly gridded MEIC emission datasets at a horizontal resolution of 4 km

× 4 km for the CMAQ model were generated by the Sparse Matrix Operator Kernel Emissions (SMOKE)

modeling system (UNC, 2014) using reasonable temporal and spatial allocation coefficients (Cheng et

al., 2017). Meteorological outputs from the WRF simulations were processed to create model-ready

inputs for CMAQ using the Meteorology–Chemistry Interface Processor (MCIP) (Otte and Pleim, 2010).

The chemical mechanism is CB05, and the boundary conditions of trace gases consist of idealized,

Northern Hemispheric, mid-latitude profiles based on results from the NOAA Agronomy Lab Regional

Oxidant Model. The model simulation was started one day before the first day of the experiment to avoid

the spin-up problem and improve the simulation accuracy.

**2.3.2 Validation of simulated surface wind and $NO_2$**

Modeled wind speeds and directions were validated by observation data from four weather stations in

Beijing. The observed hourly wind speed and direction data at the meteorological stations, shown in



Figures S2 and S3, were obtained from the China Meteorological Administration. The four stations are the Nanjiao (NJ), Tongzhou (TZ), Mentougou (MTG), and Shunyi (SY) meteorological stations, which

represent the south, east, west, and north area of Beijing, respectively. It was shown that the temporal variation in simulated wind speed at the four stations were consistent with the observations, but the simulations were higher than the observations (Fig. S2). To calculate the $E_{NOX}$ accurately, we corrected the simulated wind speed using the observation data from the four weather stations. Specifically, we computed the relative error of the modeled wind speed during every journey and then used it to correct

the simulated wind speed at all sampling points for every journey. The correlation coefficient between simulated and observed wind speeds at the four stations was 0.47, and the result passed the 99.9% significance test. The root mean square error (RMSE) was small, with a value of 1.18 m s$^{-1}$. Except for the MTG station, simulated wind directions at the other three stations were in accordance with the observations, particularly for the primary wind direction (Fig. S3). The primary wind direction and its

frequency at the MTG station were not consistent with the observations because these are affected by the complex topography near the Taihang and Yanshan mountains. Hence, the simulations of wind speed and wind direction were reliable for estimation of the $NO_X$ emissions.

Figure S4 presents the temporal variation in simulated and observed $NO_2$ concentration from January 18 to October 13, 2014. The hourly measurements of $NO_2$ concentrations (shown in Fig. S1b) were

obtained from the National Environment Monitoring Station in China. In general, the temporal variation in the $NO_2$ simulation was consistent with the observation. The simulated values were close to the observations, except for January 21–24, September 19, and October 9–10, when $NO_2$ simulations were higher than the observations. The correlation coefficient between simulated and observed $NO_2$ concentrations was 0.73, and the result passed the 99.9% significance test (Fig. S5). The RMSE and mean

absolute error (MAE) were 16.14 and 19.21 μg m$^{-3}$, respectively. Because observed $NO_2$ might include the NOz component, it can lead to a systematical biases (underestimation) of $NO_2$ by model compared to observation (Ma et al., 2012). Thus, the simulated $NO_2$ concentrations and hence the ratio of $NO_2$ and NOx were reliable for estimating $NO_X$ emissions.

## 3.    Results and Discussion

### 3.1 Tropospheric NO$_2$ VCD

Figure 4 presents the temporal variation in the tropospheric $NO_2$ VCD on the 6th Ring Rd of Beijing in

January, September, and October, 2014. In general, the $NO_2$ VCD in January was higher than that in other

months. The highest values falling between $8 \times 10^{16}$ and $13 \times 10^{16}$ molecules $cm^{-2}$ occurred on January

19, 23, and 24. The mean, maximum, and minimum $NO_2$ VCD during the sampling periods were all

larger in January than in the other two months. The mean $NO_2$ VCD ranged mostly from $4.5 \times 10^{16}$ to $9$

$\times 10^{16}$ molecules $cm^{-2}$ in January, but values were all lower than $4.5 \times 10^{16}$ molecules $cm^{-2}$ in September

and October. The $NO_2$ VCD values during the mornings of January 23 and October 13 were $9.05 \times 10^{16}$

and $1.23 \times 10^{16}$ molecules $cm^{-2}$, corresponding to the maximum and minimum values, respectively,

during the 19 circling journeys. This result may be caused by higher emissions and some meteorological

conditions that were unfavorable for dispersion and transport of pollutants in winter. Lower PBL height

and lower wind speed suppress horizontal and vertical diffusion and transport of $NO_X$. Southwest and

east winds are favorable for the transport of air pollutants from the south and east areas to the city of

Beijing. Higher cloud cover is unfavorable for photolysis of $NO_2$. A similar pattern of seasonal variation

in tropospheric $NO_2$ VCD was found previously by site MAX-DOAS measurements in Beijing(Ma et al.,

2013a;Hendrick et al., 2014).

Figure 5 shows that the monthly average $NO_2$ VCD at most sampling points on the 6th Ring Rd were

obviously higher in January than in the other two months (by a factor of two in most cases). The spatial

distribution characteristics of $NO_2$ VCD in September were similar to those in October. In addition, the

$NO_2$ VCD values at the northern and southern parts of the 6th Ring Rd were all larger than those in other

areas for all three months. The high $NO_2$ VCD in the southern region was related to strong local emissions

to the south of Beijing and transport from central and southern Hebei and the city of Tianjin (Meng et al.,

2018).. As shown in Fig. 4, the maximum journey-averaged $NO_2$ VCD occurred on the morning of

January 23, and the minimum occurred on the morning of October 13.

We investigated the spatial distribution differences in $NO_2$ VCD between these two circling journeys,

as shown in Fig. 6. The $NO_2$ VCD values on the 6th Ring Rd in the morning of January 23 were all large,

particularly in the north and southwest areas, with magnitudes of $10 \times 10^{16}$ to $12 \times 10^{16}$ molecules $cm^{-2}$.

On October 13, high $NO_2$ VCD was located in the southern areas, but values were lower in the northern

areas. The spatial distribution differences between these two journeys were related to the high emission

during the heating season in January (see section 3.2) and the impacts of the wind field. To investigate

the impact of the wind field on the spatial distribution of $NO_2$ VCD, we used thin-grid ECWMF

reanalysis data for January 23 and October 13 with a spatial resolution of $0.125° \times 0.125°$. Figure 7 shows



the distribution difference of wind field at 8:00 and 14:00 BJT on these two days, respectively. The $NO_2$

VCD was large with weak south wind and with convergence of southeast and northwest wind in Beijing

and its surrounding area, but values were far smaller with strong north wind. Weak south wind and a

breeze or calm wind resulted in the transport of $NO_2$ from the south area in Hebei Province and its

accumulation on January 23. Strong north wind suppressed the transport of $NO_2$ from the south area on

October 13. These results indicate that the wind field has large impacts on the spatial distribution of $NO_2$

VCD in Beijing.

Figure 8 presents the spatial distributions of wind and $NO_2$ VCD averaged for the three different wind

fields. The mean $NO_2$ VCD at most sampling points along the 6th Ring Rd was obviously higher under

the south wind field than under the north wind. High $NO_2$ emission in the three months was located

within the 5th Ring Rd of Beijing (Fig. 10), and the background concentrations of $NO_2$ VCD in the north

and south areas were remarkably different due to the impacts of emission sources from south areas of

Beijing, such as Hebei Province. Hence, southerly wind can transfer air pollutants from the southern area

to Beijing and lead to high $NO_2$ flux and NOx emission, whereas impacts of north wind on $NO_2$ flux and

NOx emission are smaller because the background concentration of NO2 VCD in north of Beijing is

lower. Convergence of the wind field in the south parts of the 6th Ring Rd is favorable to accumulation

of $NO_2$ from the surrounding area to the southern parts of the ring road.

**3.2 Quantification of NOx emissions**

To estimate the $NO_2$ fluxes ($F_{NO2}$) and $E_{NOX}$ accurately, we used the data from 10 circling journeys (Table

1), for which the RMSEs of simulated wind speeds at the four weather stations were all less than 1.5 m

$s^{-1}$. In addition, $NO_2$ VCD measurements at the sampling points outside of the 6th Ring Rd during 11

circling journeys were not used to quantify $F_{NO2}$ and $E_{NOX}$. Figure 9 shows the journey-to-journey

variation in estimated $F_{NO2}$ and $E_{NOX}$ over Beijing for the 10 circling journeys in January, September, and

October, 2014. The $F_{NO2}$ fell in between $1.13 \times 10^{25}$ and $11.35 \times 10^{25}$ molecules $s^{-1}$. The ranges of $E_{NOX}$

during the heating (January) and non-heating (September and October) periods were $28.7 \times 10^{25}$ to $60.0$

$\times 10^{25}$ molecules $s^{-1}$ and $7.7 \times 10^{25}$ to $24.8 \times 10^{25}$ molecules $s^{-1}$, respectively. The average $E_{NOX}$ values

in the heating and non-heating periods were $43.0 \times 10^{25}$ molecules $s^{-1}$ and $13.9 \times 10^{25}$ molecules $s^{-1}$,

respectively. In general, the journey-to-journey variation patterns of $F_{NO2}$ and $E_{NOX}$ were consistent with

that of the mean $NO_2$ VCD. In other words, the estimate of $E_{NOX}$ was determined mainly by the $NO_2$



VCD. Seasonal variation characteristics of the estimated $E_{NOX}$ were obvious. Specifically, the total $E_{NOX}$ was higher in January than in the other two months. The average $E_{NOX}$ in the heating period was about 3.1 times those in the non-heating period.

In addition to the seasonal differences, the journey-to-journey variation in estimated $E_{NOX}$ were large even within the same month, mainly due to uncertainties in the calculations of wind speed, ratio of $NO_2$ and NOx concentration, and decay rate of $NO_X$ from the emission sources to the measured positions under different meteorological conditions. In addition to the $NO_2$ VCD, wind speed, and wind direction at the sampling points, the estimated $NO_X$ emission rate is obviously affected by the Leighton ratio of

NO and $NO_2$ concentration and the lifetime of NOx (Valin, et al., 2013). Thus, the estimated $NO_X$ emission rate could be very large even if the $NO_2$ VCD was small, such as in the case of January 27. It should be noted that the mean wind speed on January 27 was relatively small and led to higher $c_\tau$, meanwhile, the ratios of $NO_X$ and $NO_2$ were relatively large, so $E_{NOX}$ on January 27 was large although $F_{NO2}$ was relatively small. Thus, if $c_\tau$ and $c_L$ are simultaneously larger, higher $E_{NOX}$ occurs. However, if

only one factor is larger and the other is smaller, such as higher $c_\tau$ and lower $c_L$ as on January 18, the morning and afternoon of September 14, and the morning of October 13, $E_{NOX}$ is lower.

**3.3 Comparisons with MEIC inventory and other estimates**

We compared the estimated $NO_X$ emission with the multi-resolution emission inventory in China (MEIC) released by Tsinghua University for 2012 (MEIC 2012) (Zhang et al., 2009;Zhang et al., 2012). The

horizontal resolution of MEIC 2012 is 0.25° × 0.25°, and five sectors, agriculture, industry, power, residents, and transportation, are included.

Figure 10 presents the spatial distributions of $NO_X$ emission rates over Beijing in January, September, and October, 2012, from MEIC. A high $NO_X$ emission zone was located within the 5th Ring Rd of Beijing, and a low emission zone occurred in other areas. The $NO_X$ emissions in January were obviously larger

than those in the other two months. The concentrated distribution of $NO_X$ emission sources within the 5th Ring Rd of Beijing indirectly indicates the applicability of Eq. (1) to estimate the $NO_X$ emission rates from the car MAX-DOAS measurements on the 6th Ring Rd in this study.

Figure 11 shows the journey-to-journey estimated $NO_x$ emission rates from car MAX-DOAS measurements in January, September, and October, 2014 (denoted as $E_{NOX}$), and the corresponding

monthly averaged $NO_x$ emission rates from the MEIC 2012 for the same region within the 6th Ring Rd





of Beijing (hereafter expressed as MEIC_Month). In most cases, the MEIC_Month was lower than the estimated $E_{NOX}$, particularly in January. The differences between the estimated $E_{NOX}$ and the MEIC_Month during some journeys were remarkably large. The differences between the $E_{NOX}$ and MEIC 2012 during the 10 journeys may be caused by (1) the interannual differences in emission inventory, (2)

the different timescales of the two emission estimates, (3) the uncertainty of the estimated $E_{NOX}$ and MEIC 2012, (4) inconsistency of wind field during the period of measurements, (5) extra transfers from source areas other than urban Beijing, and so on. Firstly, the $E_{NOX}$ in this study was estimated for the year 2014, whereas the MEIC 2012 was established for the year 2012. Secondly, our results represented only the conditions during a few measurements during daytime, whereas the MIEC 2012 denoted monthly

average conditions. Thirdly, the uncertainty of MEIC 2012 is large, particularly in autumn and winter (Li *et al*., 2017; Meng *et al*., 2018). Fourthly, the emission estimation method used in this study assumes that the wind field is constant during the period of measurements and that the wind speed is also sufficiently high that the transport across the encircled area is fast compared to the atmospheric lifetime of the trace gas (Ibrahim *et al*., 2010). However, the wind field during some journeys could have changed

systematically and been convergent or divergent in some areas of Beijing, as marked as other type of wind field in Table 1. Ibrahim *et al*. (2010) also pointed out that systematic changes during the period of measurements can become important to the emission estimate, particularly if measurements with high trace gas VCD are accompanied by strong deviations of the actual wind speed (or direction) from the assumed average values. For example, on the morning of January 27, the highest $NO_2$ VCD was measured,

and the wind field changed during the measurement journey. In such cases, the systematic changes in wind speed and direction can lead to additional uncertainties in $E_{NOX}$. Finally, because southerly wind can bring $NO_x$ emitted in the south-central regions of Hebei Province to Beijing, the $E_{NOX}$ from car MAX-DOAS measurements will be overestimated under south wind conditions.

Figure 11 also shows the uncertainty of $E_{NOX}$, calculated from the errors of measured $NO_2$ VCD,

simulated wind speed, $c_L$ and $c_\tau$ according to the error transfer formula of relative deviation. The standard deviation (STD) of wind speed over a period of time can provide a bound for the related uncertainties of the emission estimate (Ibrahim *et al*., 2010). Therefore, we first computed the uncertainty of $F_{NO2}$ based on the STD of the corrected wind speed and the measurement error of $NO_2$ VCD (about ± 10%, Ma *et al*., 2013a) for each journey. Then, we calculated STD of $c_\tau$ according to the first derivative

of equation (4) and the monthly STD of $c_L$ using its regional average data within the 6th Ring Rd of





Beijing during all journeys in each month. We used the identical STD of $c_L$ for each journey in the same month to calculate of the uncertainty of $E_{NOX}$. The results showed that the STD ranges of wind speed, $c_L$ and $c_\tau$ were 0.13–1.30 m s$^{-1}$, 0.11–0.37, and 0.17–1.97, respectively. The uncertainty range of $E_{NOX}$ was 16.4–33.2%.


### 4.     Conclusion and Discussion

We carried out 19 city-circle-around  car MAX-DOAS experiments on the 6th Ring Rd of Beijing in January, September, and October, 2014. The VCD of $NO_2$ was retrieved and the temporal and spatial distributions were investigated. Then the $NO_X$ emission rates in urban Beijing were estimated using the

measured $NO_2$ VCD together with the refined wind fields, $NO_2$ to NOx ratios, and $NO_2$ lifetimes simulated by the LAPS-WRF-CMAQ model system, and the results were compared to the emission rates from the MEIC inventory 2012.

The mean, maximum, and minimum $NO_2$ VCD values during the sampling periods in January were all larger than those in the other two months, mainly due to higher emissions in winter. The mean $NO_2$

VCD was typically larger at the southern parts of the 6th Ring Road than  northern parts because weak south wind resulted in the transport and accumulation of $NO_2$ from southern areas in Hebei Province and strong north wind suppressed the transport of $NO_2$ from the southern area. Such inhomogeneous distributions of tropospheric $NO_2$ VCD bring a challenge for validation of satellite products for Beijing as well as other megacities.

The journey-to-journey variation in estimated $E_{NOX}$ were large, even within the same month, mainly due to uncertainties in the calculation of wind speed, the ratio of $NO_2$ and NOx concentration, and the decay rate of $NO_X$ from the emission sources to the measured positions under different meteorological conditions. The ranges of $E_{NOX}$ during the heating and non-heating periods were $28.7 \times 10^{25}$ to $60.0 \times 10^{25}$ molecules s$^{-1}$ and $7.7 \times 10^{25}$ to $24.8 \times 10^{25}$ molecules s$^{-1}$, respectively. The average $E_{NOX}$ values in

the heating and non-heating periods were $43.0 \times 10^{25}$ molecules s$^{-1}$ and $13.9 \times 10^{25}$ molecules s$^{-1}$, respectively. The uncertainty range of $E_{NOX}$ was 16.4–33.2%. The monthly emission rates in the area within the 6th Ring Rd of Beijing from MEIC 2012 were lower than the estimated $E_{NOX}$, particularly in January. The differences between the $E_{NOX}$ and the monthly emission rates from MEIC 2012 may be attributable to the interannual differences in the emissions inventory, the different timescales and

uncertainties of two kinds of inventory, inconsistencies of wind field during the period of measurements,



and extra transfers from source areas other than urban Beijing.

Our results showed that car MAX-DOAS measurements can be used effectively for dynamic monitoring and updating of the $NO_x$ emissions from megacities such as Beijing. To estimate $E_{NOX}$ by car MAX-DOAS accurately in Beijing and other similar megacities, appropriate meteorological conditions, such as smaller fluctuations of the wind field, relatively larger wind speed, and suitable wind direction, need to be selected to avoid the impact of extra transfers of large emission sources from surrounding areas. In addition to the $NO_2$ VCD, simultaneous observations of wind speed, wind direction, and surface NO and $NO_2$ concentrations are recommended to reduce the uncertainties of $c_\tau$ and $c_L$.

*Data availability.* The NCEP-FNL reanalysis and ECMWF are publicly available at http://rda.ucar.edu/datasets/ds083.2/ and https://www.ecmwf.int/en/forecasts/datasets , respectively. The $NO_2$ - measurements and meteorological observations including wind speed and wind direction data are available at http://113.108.142.147:20035/emcpublish and http://data.cma.cn/ , respectively. The tropospheric $NO_2$ VCD data derived from this study are available on the request.


*Author contributions.* JM and XC designed the research. JM, JJ, JG, MQ, QX, and PY contributed to the measurements, and JM performed the spectral analysis and retrieval. XC and JP designed the model experiment and performed the model simulations. XC, YL, JP, and XM contributed to the data processing and analyses. XC and JM analyzed the results and wrote the paper with inputs from all authors.


*Competing interests.* The authors declare that they have no conflicts of interest.

*Acknowledgments.* This work was supported jointly by the National Natural Science Foundation of China (91644223), the National Research Program for Key Issues in Air Pollution Control (DQGG0104), and the Scientific and Technological Development Funds from the Chinese Academy of Meteorological Sciences (2018KJ042). The authors acknowledge Tsinghua University for providing the emissions inventory and the China National Environmental Monitoring Centre for providing surface $PM_{2.5}$ observation data.



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





Table 1. Sampling periods of the car MAX-DOAS experiment and corresponding meteorological conditions over Beijing in January, September, and October, 2014.

| Journey | Date | Time (BJT) | Wind speed (m/s) | Type of wind field* | Total cloud fraction | PBL Height (m) |
|---|---|---|---|---|---|---|
| 1** | 2014/1/18 | 10:48-13:09 | 2 | O | 0 | 564 |
| 2 | 2014/1/19 | 13:31-15:40 | 1 | O | 7 | 167 |
| 3 | 2014/1/21 | 13:15-15:32 | 3 | S | 0 | 163 |
| 4 | 2014/1/23 | 10:39-12:25 | 3 | O | 7 | 187 |
| 5 | 2014/1/23 | 13:07-15:12 | 2 | O | 7 | 163 |
| 6 | 2014/1/24 | 10:42-12:03 | 2 | N | 8 | 39 |
| 7 | 2014/1/24 | 13:03-15:09 | 3 | N | 8 | 39 |
| 8** | 2014/1/26 | 10:21-12:13 | 5 | S | 5 | 341 |
| 9** | 2014/1/27 | 09:11-11:38 | 2 | O | 7 | 75 |
| 10** | 2014/1/27 | 13:30-15:28 | 2 | O | 0 | 178 |
| 11** | 2014/9/14 | 09:40-12:52 | 4 | N | 10 | 173 |
| 12** | 2014/9/14 | 15:02-17:17 | 2 | N | 10 | 226 |
| 13** | 2014/9/17 | 09:07-11:42 | 2 | O | 7 | 173 |
| 14 | 2014/9/19 | 09:09-11:50 | 2 | S | 3 | 178 |
| 15 | 2014/10/9 | 13:04-14:44 | 1 | S | 7 | 43 |
| 16 | 2014/10/10 | 09:52-12:28 | 2 | S | 7 | 663 |
| 17** | 2014/10/12 | 14:02-16:42 | 3 | N | 7 | 167 |
| 18** | 2014/10/13 | 09:12-11:59 | 3 | O | 0 | 186 |
| 19** | 2014/10/13 | 13:11-16:27 | 3 | O | 0 | 130 |

*Three types of wind filed are South (S), North (N) and Other (O).

**The data from ten circling journeys are used to estimate the NOx emission.








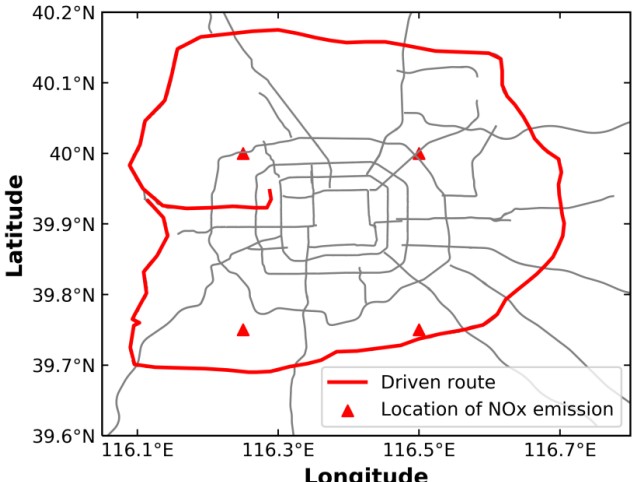

**Fig. 1** Driving routes (red line) of the car MAX-DOAS experiment on the 6th Ring Rd of Beijing.



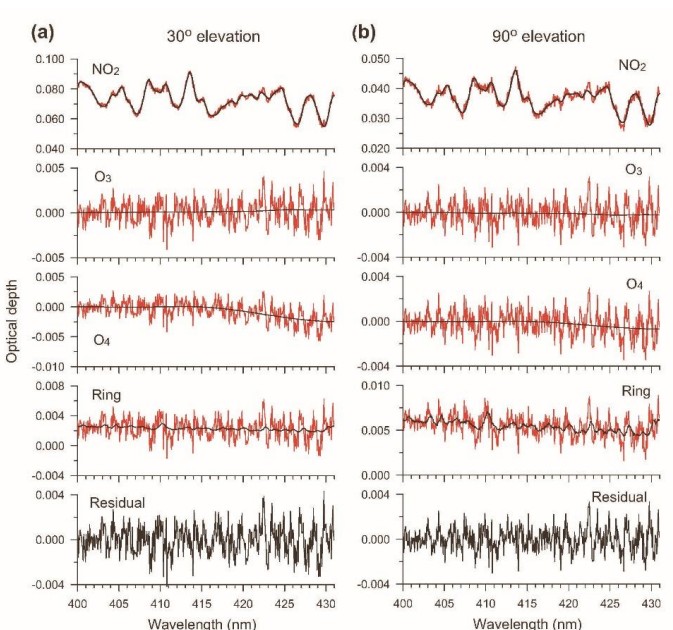

**Fig. 2** Examples of the NO$_2$ retrieval from two successive spectra measured (**a**) at a 30° elevation angle
(with NO$_2$ differential slant column density (DSCD) of $1.23 \times 10^{17}$ molecules cm$^{-2}$) and (**b**) at a 90°
elevation angle (with NO$_2$ DSCD of $6.22 \times 10^{16}$ molecules cm$^{-2}$) on January 18, 2014, at around 11:40
BJT.








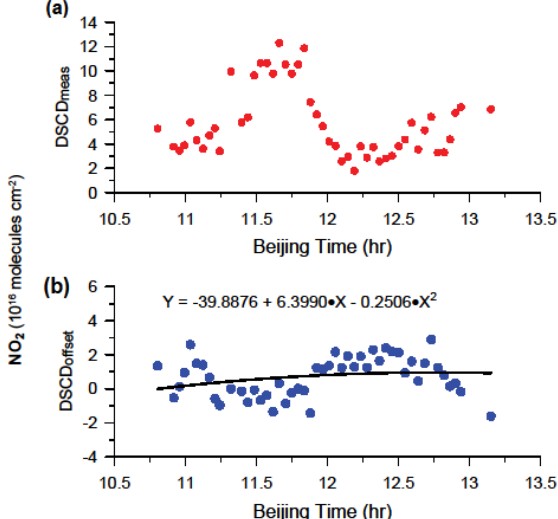

**Fig. 3** Time series of the $NO_2$ (**a**) $DSCD_{means}$ (red dots) and (**b**) $DSCD_{offset}$ (black dots) (units of $10^{16}$ molecules $cm^{-2}$) for the 30° elevation angle of each sequence on January 18, 2014. The black curve represents a second-order polynomial fit from individual $DSCD_{offset}$ data points.







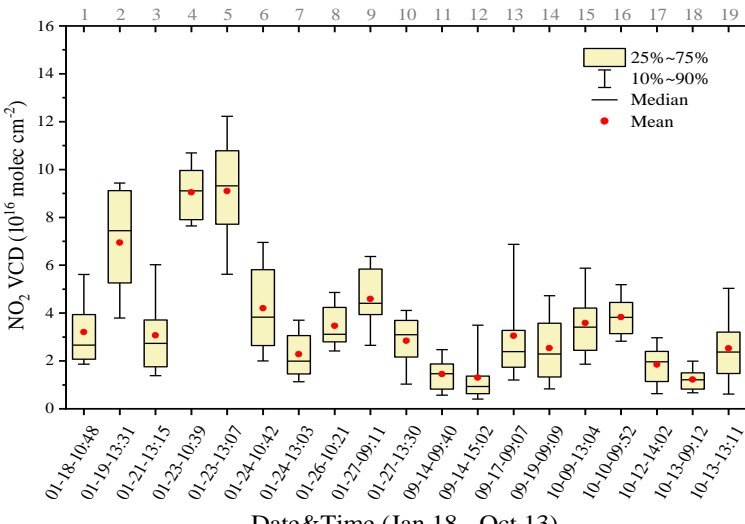

**Fig. 4** Time series of the tropospheric $NO_2$ vertical column density (VCD) for 19 circling journeys on

the Sixth Ring Road of Beijing in January, September, and October, 2014. Lower (upper) error bars and

yellow boxes are the 10th (90th) and 25th (75th) percentiles of the data of each journey, respectively.

Hyphens inside the boxes are the medians, and red circles are the mean values. The numbers of each

journey are labeled at the top axis. See Table 1 for detailed information about each journey.





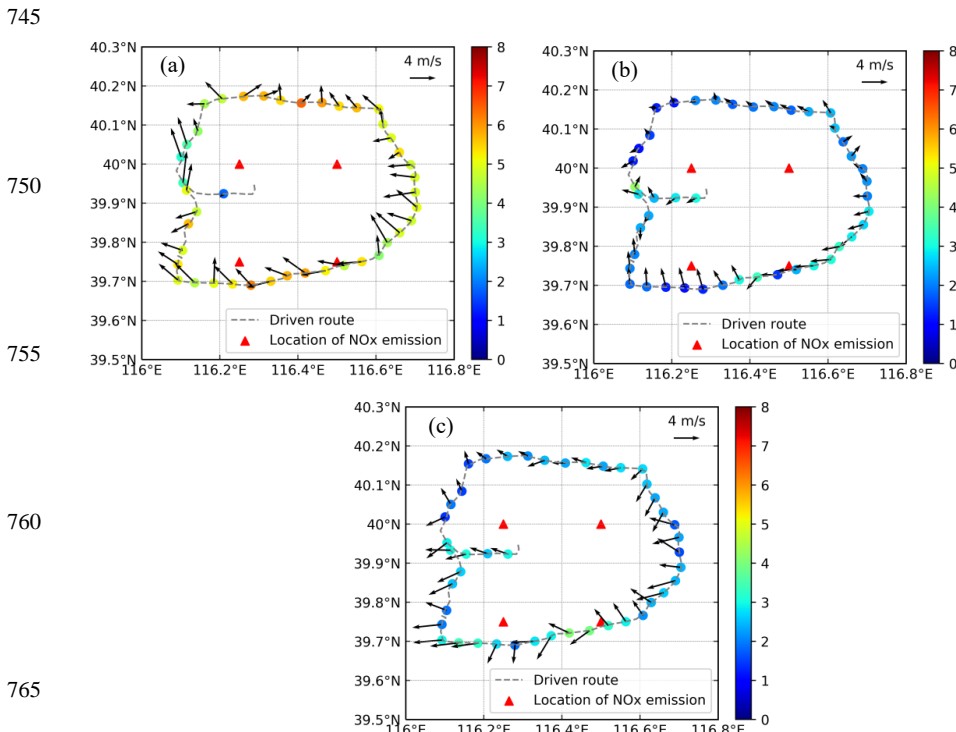

**Fig. 5** Distributions of the monthly average $NO_2$ VCD (E16 molecules $cm^{-2}$) on the 6th Ring Rd of

Beijing in **(a)** January, **(b)** September, and **(c)** October, 2014.



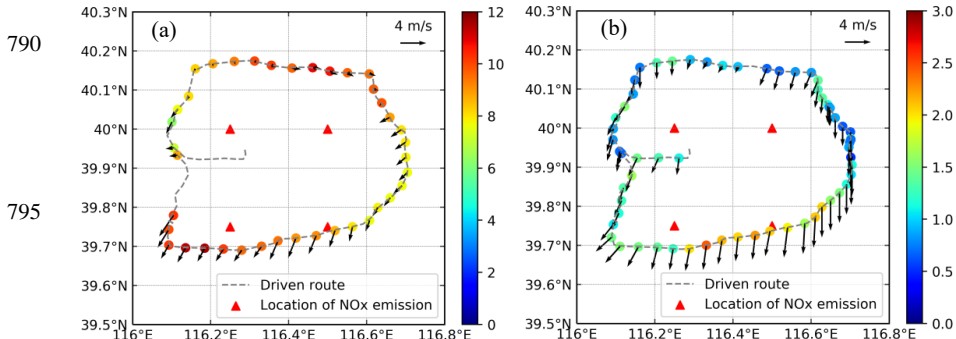

**Fig. 6** Distributions of the maximum and minimum NO$_2$ VCD (E16 molecules cm$^{-2}$) on the 6th Ring Rd of Beijing on the morning of **(a)** January 23 and **(b)** October 13, 2014.




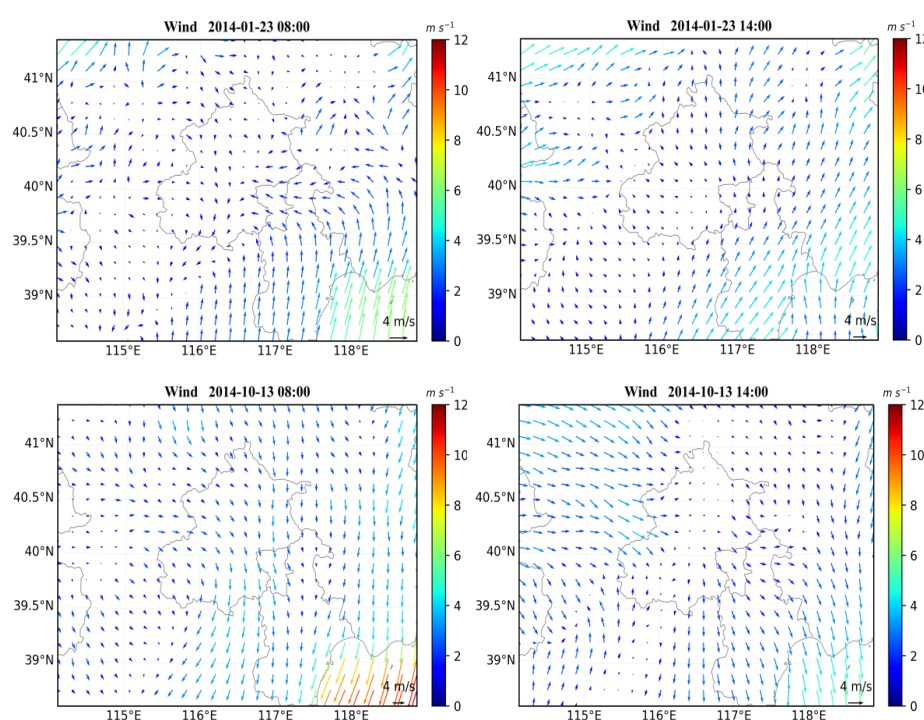

**Fig. 7** Wind fields in Beijing and the surrounding area from ECWRF at 08:00 (left column) and 14:00

(right column) BJT on January 23 and October 13, 2014.







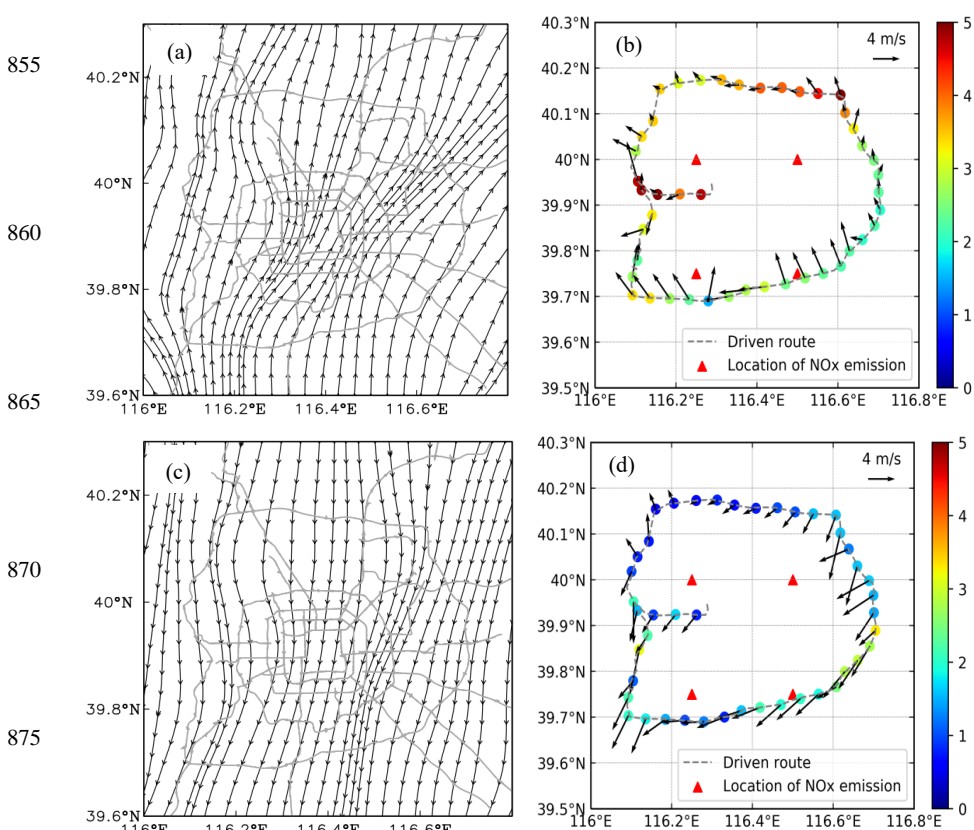






**Fig. 8** Average wind stream and NO$_2$ VCD (E16 molecules cm$^{-2}$) distributions under the three different

types of wind field over Beijing: **(a)** south wind, **(b)** NO$_2$ VCD under south wind, **(c)** north wind, and **(d)**

NO$_2$ VCD under north wind.












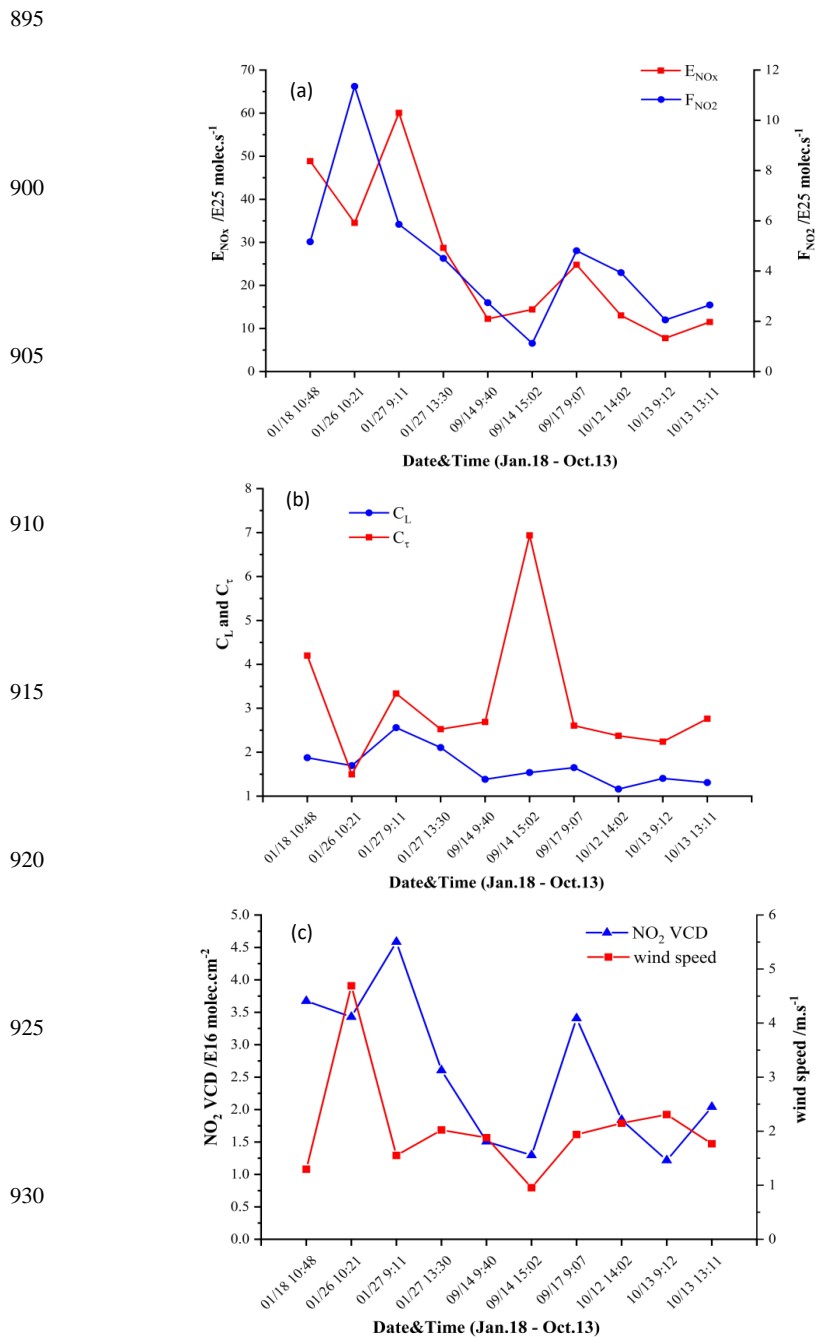

**Fig. 9** Journey-to-journey variation in **(a)** $F_{NO_2}$ and $E_{NO_x}$, **(b)** $c_\tau$ and $c_L$, **(c)** NO$_2$ VCD and mean wind

speed for 10 circling journeys on the 6th Ring Rd of Beijing in January, September, and October, 2014.






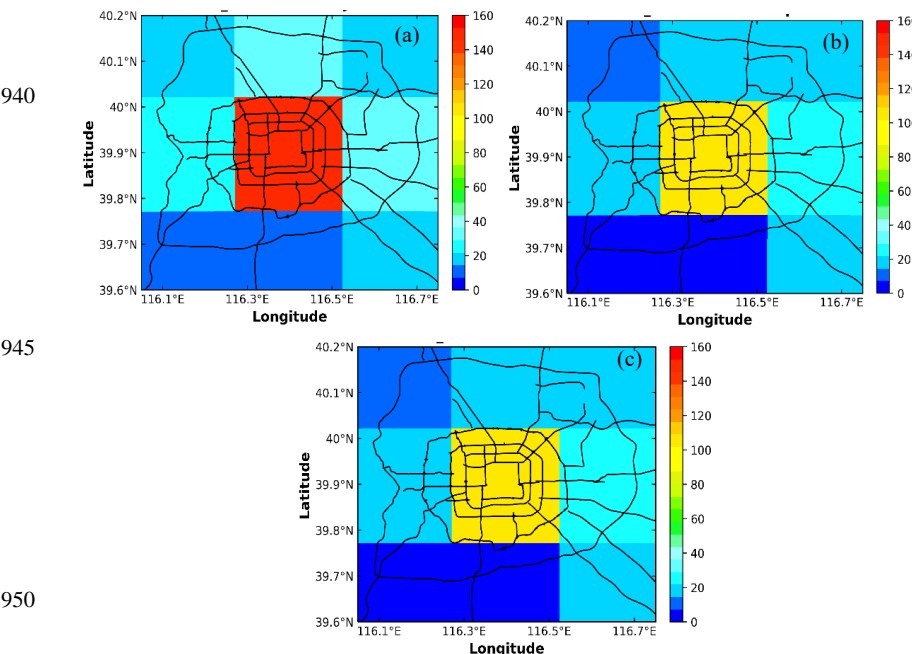

**Fig. 10** Spatial distributions of NO$_X$ emissions (mole km$^{-2}$ h$^{-1}$) over Beijing based on the MEIC inventory

in **(a)** January, **(b)** September, and **(c)** October 2012.










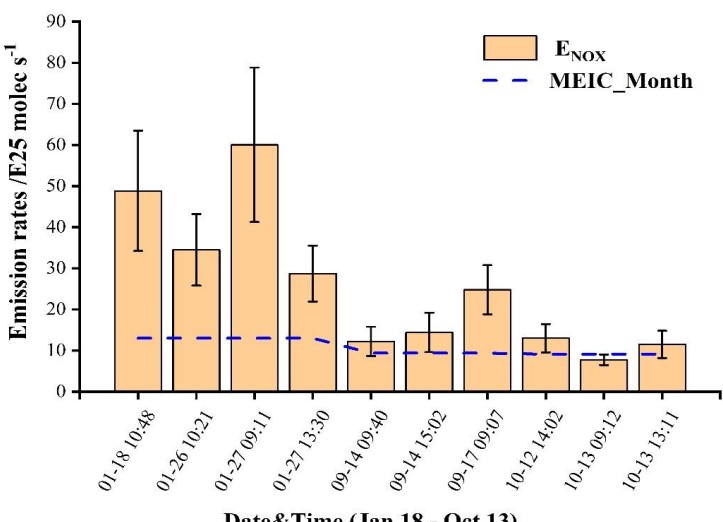

**Fig. 11** Journey-to-journey variation in estimated $E_{NO_x}$ and corresponding monthly emissions rates from the MEIC inventory (MEIC_Month) within the 6th Ring Rd of Beijing in January, September, and October 2014. Error bars represent the uncertainties in estimated $E_{NO_x}$




