# Peer review of "Fig.1S Triple-nested domains of (a) the LAPS-WRF-CMAQ model system and (b) the distribution of meteorological/environmental monitoring stations."

_Atmospheric Chemistry and Physics, 2020_

## Referee Comment (RC1) · Anonymous Referee #2 · 17 Jul 2020

General Remarks:

The manuscript by Cheng et al. describes mobile-MAX-DOAS measurements of NO2 VCDs in the Beijing area. The study is based on 19 circuits around the 6th Ring Road in Beijing during two seasons in 2014. Emissions of NOx by Beijing were estimated using the MAX-DOAS VCDs as well as winds and NOx/NO2 ratios simulated by the LAPS-WRF-CMAQ model system. Simulated winds were validated using wind observations at multiple locations in the region. The seasonal differences between average VCDs and NOx emissions were compared and found to be greater during the heating periods.

[Figure]

The estimated NOx emissions were compared to a bottom-up emission inventory from 2012.

The manuscript illustrates how mobile-MAX-DOAS measurements in Beijing and other megacities could provide dynamic monitoring of NOx emissions and validation of satellite NO2 VCDs. My concerns are about the application and interpretation of statistics for the validation of modelled winds (Issue #1), and about the validity of the scientific methods and assumptions for some of the measurement periods and in the calculation of errors (Issue #2), and error calculation (Issue #3). The authors should address the following specific issues and minor corrections before the manuscript is published.

Specific Issues:

Issue #1) In general, be careful about application and interpretation of statistics for discussion of means and validation of wind data. Some descriptions of error calculations are not sufficiently complete to allow their reproduction.

-Page 2 line 32-33: please include an uncertainty value when stating mean values throughout and, preferably, indicated the statistical type of uncertainty (e.g., standard deviation of mean).

-Page 10 line 266-267: what about Fig. S2 shows that the temporal variation between the simulated and observed wind speed were consistent? An R of 0.47 is quite low given that the R2= 0.22, meaning only 22% of variation in the model is explained by variation in the observation. In terms of the simulated wind-speed being greater than the observations, if that conclusion is from the slope being >1, the slope should be assessed with an y-intercept forced to zero. In an ideal case the simulated and observed speeds would be the same (1:1 line, 0 intercept) unless a systematic error or offset is expected. Also, the slope is barely above zero. Using a linear fitting algorithm program that provides the expected error on the slope and intercept will indicate whether the slope was statistically greater than zero.

-Page 10 line 268-269: it is still unclear how the relative error was used to "correct" the simulated winds. Did you simply add it as an error bar?

-Page 10 Page 273-274 The wind rose plots (Fig. S3) give little indication that the wind-direction was consistent between the simulation and observations during you driving periods. Without this information, the claim that the modelled winds were accurate is unsupported. Either apply circular statistics to determine the R2 of simulated and observed wind-directions or provide time series plots of wind-direction (perhaps in the supplemental).

-Page 10 lines 283-284 The regression statistics should be determined using a linear correlation line with an intercept forced to zero (see Page 10 line 266-267 comment).

Issue #2) A major assumption of the mobile-MAX-DOAS method for estimating emissions using a flux integral is that winds stayed relatively constant during the measurement period. Therefore, this method is only valid to use on MAX-DOAS data with these wind conditions.

-Page 14 line 386, point (4): the driving routes should be screened for changing wind field since relatively constant wind fields are a fundamental assumption of Eq. (1) for mobile-MAX-DOAS and, if violated, lead to large, unquantified uncertainties. This point is stated in the manuscript Page 16 lines 444-446 but contrasts with the inclusion of emission estimates in your figures and comparisons where large wind fluctuations occurred It is not scientifically valid to use Eq. (1) under these conditions and the emission estimates from these routes should not be presented unless designated as having unknown error.

-Page 14 line 395: Therefore, exclude journeys of wind type "other". See comment above.

Issues #3)

-Page 14 line 408-409 The total error values appear too small given all the potential

factors contributing to uncertainty. Shaiganfar et al. (2011) suggests that the error of using the VCDgeo calculation alone is up to 20% (https://doi.org/10.5194/acp-11-10871-2011). A VCD error 10% is likely too small if both the SCD retrieval error and the VCDgeo errors are combined. Was the error on the simulated wind speed the RMSE between the modelled and observed winds during the driving period? What about the contribution of wind-direction error? If the wind-direction varied a lot, the error contribution would likely dominate.

Minor Issues and Corrections:

Page 2 line 41: replace "consumption" of fossil fuels with "combustion".

Page 2 line 47: remove the word "obviously" here and throughout as it is unnecessary.

Page 2 lines 54-56: since you begin with a discussion of PM, may want to explicitly describe that NOx is a precursor of nitric acid, which is a precursor of nitrate aerosols for maximum clarity.

Page 2 line 60: replace "high" with "large".

Page 3 line 64: define "top-down constraint" or use "top-down" emission estimate

Page 3 line 71: there is a missing space between sentences.

Page 3 line 72: specify whether it is a large decrease in precision or accuracy?

Page 3 line 74-75: please add citation(s) for these uncertainty factors.

Page 3 line 80 MAX-DOAS has been around for nearly two decades (see below).

Honninger, G. and Platt, U.: Observations of BrO and its vertical distribution during surface ozone depletion at Alert, Atmos. Environ., 36, 2481–2489, https://doi.org/10.1016/S1352-2310(02)00104-8, 2002.

Hönninger, G., von Friedeburg, C., and Platt, U.: Multi axis differential optical absorption spectroscopy (MAX-DOAS), Atmos. Chem. Phys., 4, 231–254,

https://doi.org/10.5194/acp-4-231-2004, 2004.

Wagner, T., Dix, B., von Friedeburg, C., Friess, U., Sanghavi, S., Sinreich, R., and Platt, U.: MAX-DOAS O4 measurements: A new technique to derive information on atmospheric aerosols – Principles and information content, J. Geophys. Res.-Atmos.,109, D22205, https://doi.org/10.1029/2004JD004904, 2004.

Page 3 line 82: add "pointing" between "zenith" and "directions"

Page 3 line 82: add "vertical" before "profiles"

Page 3 lines 84-86: I suggest adding these citations of MAX-DOAS NO2 measurements (below)

Tan et al. (2018) doi:10.5194/acp-18-15387-2018

Wagner et al. (2011) doi:10.5194/amt-4-2685-2011

Page 4 line 91: add "horizontal" before "spatial distribution of pollutants" (since the stationary inverse modelling MAX-DOAS technique also gives spatial information but in the vertical).

Page 4 line 95: in general, you may want to mention that mobile-MAX-DOAS could be very useful for validating the NO2 VCDs and NOx emission estimates from the new, high pixel resolution measurements by the TROPOMI instrument on the Sentinel-5P satellite.

Page 4 line 105: Consider adding the study objectives or aims before listing the sections. It can help the reader quickly determine if the paper of interest and makes it easier to follow the major conclusions.

Page 4 line 115: Please explain why wind-speeds at 10 m above surface was used instead of averaged within the boundary layer height from the model output given can be well-mixed into the boundary layer (up to ~600 m in your study) at the location point and, thus, the 10 m wind speed is a lower-bound.

Page 5 line 142: The word "mounted" is more appropriate than "settled".

Page 5 line 143: Please provide information about the instrument specifications (e.g., spectrometer model, spectral resolution, cooling mechanism etc.)

Page 6 line: How many measured spectra were averaged to produce a single measurement spectrum?

Page 5 line 145: This MAX-DOAS instrument was also used more recently in the Davis et al. (2019) mobile-MAX-DOAS measurements of NOx emissions.

Page 6 line 155: Please specify whether the sequence measured multiple 30o spectra or if every 30o measurement was immediately followed by a 90o measurement.

Page 6 line 164: Please quantify "changed slightly". Example, the wind changed by < X degrees and < Y m/s during the circle journey.

Page 6 line 175: add "absorption" in front of cross sections.

Page 6 line 176 typo: "dimmer" should be "dimer"

Page 7 line 185: Do you mean "for in-situ MAX-DOAS measurements"?

Page 7 line 187: Should this be "extending Eq. (6) to Eq. (7)" ?

Page 7 line 194: replace "site" typo with "in-situ".

Page 7 line 196: check equation number.

Page 7 Line 197: add (SZA) to DSCD_offset.

Page 7 line 204: check the Eq. number.

Page 7 line 205: replace "geometry" with "geometric" and cite Brinksma et al, 2008 https://doi.org/10.1029/2007JD008808 and Wagner et al. 2010 https://doi.org/10.5194/amt-3-129-2010

Page 8 line 209-210: the meaning of this sentence is unclear.

Page 8 line 211: NO2 needs subscript.

Page 8 line 215: missing space after "September 23".

Page 8 line 216: missing period at the end of this sentence.

Page 9 line 233: hyphen needed between "three" and "dimensional"

Page 9 line 251: delete the extra period.

Page 10 line 265: change "area" to "areas"

Page 11 line 294: rewrite the sentence to read "The highest values were between…"

Page 11 line 296: see comment for Page 2 line 32-33.

Page 11 line 300-301: what factors would increase emissions in January compared to October? More home heating?

Page 11 line 301-302: were lower PBL heights and smaller wind-speeds found in January compared to September/October?

Page 12 line 337: subscript needed on NO2 VCD.

Page 12 line 349: see comment for Page 2 line 32-33.

Page 13 line 371: considering replacing "residents" with "residential"

Page 13 lines 375-377: in what way does it indicate the applicability? Please explain/elaborate. Page 22 623: fix error in reference name.

Pages 28 and 29: Figures 5 and 6: correct the unit notation (x1016) of VCDs in the caption.

Page 31 Figure 8: the caption says "under three types of wind fields" but only south and north winds are shown.

---

## Referee Comment (RC2) · Anonymous Referee #1 · 19 Jul 2020

General Remarks: The manuscript present the results of observing NO2 emission measurements in Beijing based on the Car MAX-DOAS technology. Through 19 times of city-circle-around Car-MAX-DOAS experiments, the author showed the potential of Car MAX-DOAS measurement technology in atmospheric monitoring. This observation method can be effectively used for dynamic monitoring of urban NO2 emissions. However, the database the authors use for the conclusions is relatively weak. So some revisions are needed to consider this manuscript for publication in ACP. Major concerns: 1. Line 24-25, "typically larger NO2 VCD at the southern parts of the 6th Ring

Road than at the northern parts". According to Figures 5 and 6, the NO2 VCD at the southern parts and northern parts were not typically different in January. 2. Since each measurement time was different, from 2 to 2.5 hours (sometimes nearly 3 hours), the author should introduce the traffic situation during the measurement and analyze the impact on the measurement results. 3. The author should introduce the NOx emission sources in Figure 1 and analyze the influence on the measurement. 4. The results of Car Max- DOAS measurement show that NOx emission in heating season is nearly three times as much as that in non heating season, which is obviously higher than that calculated by MEIC inventory estimate. Since central heating is adopted in Beijing urban area, the author should analyze heating season NOx sources in detail, and evaluate the contributions to the measurement results. Minor comments: 1. Abstract, here it is more appropriate to use "different months" instead of "different seasons". 2. Line 43, "less than" instead of "smaller than" 3. Line 54-56, "NO and NO2 (together denoted as NOX) form primarily in combustion processes, and the conversion between NO and NO2 in the atmosphere is very rapid" is well known and meaningless here 4. Please unify the format of "Car-MAX-DOAS" in the manuscript. For example: line 90 "car-MAX-DOAS", line 94 "Car-MAX-DOAS", line 101 "car MAX-DOAS", line 190 "car MAX-DOAS" et al. 5. Line 147, "the roof of a car" instead of "the roof a car".

---

## Author Comment (AC1) · 17 Aug 2020

**Response to Interactive comments from Anonymous Referee #2**

Referee comments are in black. Author responses are in blue.

General Remarks:
The manuscript by Cheng et al. describes mobile-MAX-DOAS measurements of NO2 VCDs in the Beijing area. The study is based on 19 circuits around the 6th Ring Road in Beijing during two seasons in 2014. Emissions of NOx by Beijing were estimated using the MAX-DOAS VCDs as well as winds and NOx/NO2 ratios simulated by the LAPSWRF-CMAQ model system. Simulated winds were validated using wind observations at multiple locations in the region. The seasonal differences between average VCDs and NOx emissions were compared and found to be greater during the heating periods. The estimated NOx emissions were compared to a bottom-up emission inventory from 2012. The manuscript illustrates how mobile-MAX-DOAS measurements in Beijing and other megacities could provide dynamic monitoring of NOx emissions and validation of satellite NO2 VCDs. My concerns are about the application and interpretation of statistics for the validation of modelled winds (Issue #1), and about the validity of the scientific methods and assumptions for some of the measurement periods and in the calculation of errors (Issue #2), and error calculation (Issue #3). The authors should address the following specific issues and minor corrections before the manuscript is published.

We thank the anonymous referee for his/her insightful and constructive comments. Below are our point-to-point responses in detail.

Specific Issues:
Issue #1) In general, be careful about application and interpretation of statistics for discussion of means and validation of wind data. Some descriptions of error calculations are not sufficiently complete to allow their reproduction.

Thanks for pointing out this. We have deleted the inappropriate interpretation of statistics for the validation of modelled winds in the revised manuscript.

-Page 2 line 32-33: please include an uncertainty value when stating mean values throughout and, preferably, indicated the statistical type of uncertainty (e.g., standard deviation of mean).

We have added the standard deviation in lines 34-35 of the revised manuscript.

-Page 10 line 266-267: what about Fig. S2 shows that the temporal variation between the simulated and observed wind speed were consistent? An R of 0.47 is quite low given that the R2= 0.22, meaning only 22% of variation in the model is explained by variation in the observation. In terms of the simulated wind-speed being greater than the observations, if that conclusion is from the slope being >1, the slope should be assessed with an y-intercept forced to zero. In an ideal case the simulated and observed speeds would be the same (1:1 line, 0 intercept) unless a systematic error or offset is expected.

Also, the slope is barely above zero. Using a linear fitting algorithm program that provides the expected error on the slope and intercept will indicate whether the slope was statistically greater than zero.

We have added a plot showing the time serial of wind speed to the revised supplement (see Fig.4s) and the temporal variation between the simulated and observed wind speed are largely consistent. Although the R of 0.47 is low and there are systematic errors in the modelled wind speed due to impacts of the complex topography and limited observation data assimilated to the LAPS-WRF model, we reduced the error of the modelled wind speed by correction based on measurements at four weather stations in Beijing. In addition, we updated Fig.S2 with an y-intercept forced to zero, and the expected error (standard deviation) of the re-calculated slope calculated with the linear fitting algorithm program is 0.002.

-Page 10 line 268-269: it is still unclear how the relative error was used to "correct" the simulated winds. Did you simply add it as an error bar?

Yes, we added the relative error bar to the simulated wind speed at sampling position of Car MAX-DOAS experiments for each journey. The corresponding descriptions can be seen in lines 318-322 of the revised manuscript.

-Page 10 Page 273-274 The wind rose plots (Fig. S3) give little indication that the wind direction was consistent between the simulation and observations during you driving periods. Without this information, the claim that the modelled winds were accurate is unsupported. Either apply circular statistics to determine the R2 of simulated and observed wind-directions or provide time series plots of wind-direction (perhaps in the supplemental).

We have given the time series of wind-direction in Fig.S4 in the revised supplement and corresponding description in lines 328-331 of the revised manuscript.

-Page 10 lines 283-284 The regression statistics should be determined using a linear correlation line with an intercept forced to zero (see Page 10 line 266-267 comment).

Done.

Issue #2) A major assumption of the mobile-MAX-DOAS method for estimating emissions using a flux integral is that winds stayed relatively constant during the measurement period. Therefore, this method is only valid to use on MAX-DOAS data with these wind conditions.

We have skipped over the journeys with the inappropriate wind data and given the error contribution of wind speed and direction to the uncertainty of estimated NOx emissions in the revised manuscript.

-Page 14 line 386, point (4): the driving routes should be screened for changing wind field since relatively constant wind fields are a fundamental assumption of Eq. (1) for mobile-MAX-DOAS and, if violated, lead to large, unquantified uncertainties. This point is stated in the manuscript Page 16 lines 444-446 but contrasts with the inclusion

of emission estimates in your figures and comparisons where large wind fluctuations occurred. It is not scientifically valid to use Eq. (1) under these conditions and the emission estimates from these routes should not be presented unless designated as having unknown error.

We excluded the journeys of wind type "O" and given detailed descriptions about the selection process of appropriate wind data for the NOx emission estimation in lines 206-218 and lines 348-359 in the revised manuscript. In addition, we have given the error contribution of wind speed and direction to the uncertainty of estimated NOx emissions in lines 519-532 of the revised manuscript.

-Page 14 line 395: Therefore, exclude journeys of wind type "other". See comment above.

Done.

Issues #3)

-Page 14 line 408-409 The total error values appear too small given all the potential factors contributing to uncertainty. Shaiganfar et al. (2011) suggests that the error of using the VCDgeo calculation alone is up to 20% (https://doi.org/10.5194/acp-11-10871-2011). A VCD error 10% is likely too small if both the SCD retrieval error and the VCDgeo errors are combined. Was the error on the simulated wind speed the RMSE between the modelled and observed winds during the driving period? What about the contribution of wind-direction error? If the wind-direction varied a lot, the error contribution would likely dominate.

We re-calculated the $NO_X$ emissions and its uncertainties using vertical average data of wind speed and direction, the ratio of $NO_X$ and $NO_2$, and the NOx lifetime from surface to 1000m at every sampling position on the 6th Ring Rd of Beijing for each journey, and provided updated results of $E_{NOX}$ and corresponding discussions in lines 135-139, lines 158-163, lines 428-437, and lines 503-532 in the revised manuscript. The updated range of emission errors of NOx emission is 19.52–52.01%. A VCD error 10% was suggested in the paper of Ma et al., 2013 and used to calculate the uncertainty of $E_{NOX}$, which seems to be a low limit of VCD errors. We have used the RMSE between the modelled and observed wind speed during the driving period to select appropriate wind data to update the NOx emission estimations and adopted the standard deviation to calculate error contributions of wind speed to the uncertainty of $E_{NOX}$. We have also calculated the contribution of wind-direction error to the estimation of $E_{NOX}$ in Table 3 and given corresponding discussions in lines 462-469 of page 16 in the revised manuscript. The results show that the error contributions from the wind field dominate, as you expected.

Minor Issues and Corrections:

Page 2 line 41: replace "consumption" of fossil fuels with "combustion".

Done.

Page 2 line 47: remove the word "obviously" here and throughout as it is unnecessary.

Done.

Page 2 lines 54-56: since you begin with a discussion of PM, may want to explicitly describe that NOx is a precursor of nitric acid, which is a precursor of nitrate aerosols for maximum clarity.
Done.

Page 2 line 60: replace "high" with "large".
Done.

Page 3 line 64: define "top-down constraint" or use "top-down" emission estimate
We have used the latter.

Page 3 line 71: there is a missing space between sentences.
Corrected.

Page 3 line 72: specify whether it is a large decrease in precision or accuracy?
It is accuracy, incorporated in the revised version.

Page 3 line 74-75: please add citation(s) for these uncertainty factors.
We have added four citations in line 83-84 of the revised version.

Page 3 line 80 MAX-DOAS has been around for nearly    two decades (see below).
Honninger, G. and Platt, U.: Observations of BrO and its vertical distribution during surface ozone depletion at Alert, Atmos. Environ., 36, 2481–2489, https://doi.org/10.1016/S1352-2310(02)00104-8, 2002.
Hönninger, G., von Friedeburg, C., and Platt, U.: Multi axis differential optical absorption spectroscopy (MAX-DOAS), Atmos. Chem. Phys., 4, 231–254, https://doi.org/10.5194/acp-4-231-2004, 2004.
Wagner, T., Dix, B., von Friedeburg, C., Friess, U., Sanghavi, S., Sinreich, R., and Platt, U.: MAX-DOAS O4 measurements: A new technique to derive information on atmospheric aerosols – Principles and information content, J. Geophys. Res.-Atmos.,109, D22205, https://doi.org/10.1029/2004JD004904, 2004.
We have replaced "the last decade" with "the last two decades" in line 90 of the revised version. All these references have been included in the revised manuscript.

Page 3 line 82: add "pointing" between "zenith" and "directions"
Done.

Page 3 line 82: add "vertical" before "profiles"
Done.

Page 3 lines 84-86: I suggest adding these citations of MAX-DOAS NO2 measurements (below)

Tan et al. (2018) doi:10.5194/acp-18-15387-2018
Wagner et al. (2011) doi:10.5194/amt-4-2685-2011
Done.

Page 4 line 91: add "horizontal" before "spatial distribution of pollutants" (since the stationary inverse modelling MAX-DOAS technique also gives spatial information but in the vertical).
Done.

Page 4 line 95: in general, you may want to mention that mobile-MAX-DOAS could be very useful for validating the NO2 VCDs and NOx emission estimates from the new, high pixel resolution measurements by the TROPOMI instrument on the Sentinel-5P satellite.
We have added "and validating the NO2 VCDs and NOx emission estimates from the new, high pixel resolution measurements by the TROPOMI instrument on the Sentinel-5P" in lines 101-103 of the revised version.

Page 4 line 105: Consider adding the study objectives or aims before listing the sections. It can help the reader quickly determine if the paper of interest and makes it easier to follow the major conclusions.
We have added the study objectives in lines 115-118 of the revised version.

Page 4 line 115: Please explain why wind-speeds at 10 m above surface was used instead of averaged within the boundary layer height from the model output given can be well-mixed into the boundary layer (up to 600 m in your study) at the location point and, thus, the 10 m wind speed is a lower-bound.
We have re-calculated the $NO_X$ emissions and its uncertainties using vertical average data of wind speed and direction from surface to 1000m altitude.

Page 5 line 142: The word "mounted" is more appropriate than "settled".
Corrected.

Page 5 line 143: Please provide information about the instrument specifications (e.g., spectrometer model, spectral resolution, cooling mechanism etc.)
We add the related information in lines 176-183 of page 6-7 in the revised manuscript.

Page 6 line: How many measured spectra were averaged to produce a single measurement spectrum?
There were about 200-400 scans typically for each single measurement spectrum. This information has been added to the revised manuscript.

Page 5 line 145: This MAX-DOAS instrument was also used more recently in the Davis et al. (2019) mobile-MAX-DOAS measurements of NOx emissions.
We have added the citation in line 174 of the revised version.

Page 6 line 155: Please specify whether the sequence measured multiple 30o spectra or if every 30o measurement was immediately followed by a 90o measurement.
Every 30° measurement was immediately followed by a 90° measurement, and this information has been added to line 191-193 of the revised version.

Page 6 line 164: Please quantify "changed slightly". Example, the wind changed by <X degrees and < Y m/s during the circle journey.
The description is not appropriate and we have deleted it (lines 202-303 of the revised version). Large wind variability occurred in some journeys and we have added the wind changes of every journey in Table 2.

Page 6 line 175: add "absorption" in front of cross sections.
Done.

Page 6 line 176 typo: "dimmer" should be "dimer"
Corrected.

Page 7 line 185: Do you mean "for in-situ MAX-DOAS measurements"?
Yes, we replace "site" with "in-situ" in line 236 and 245 of the revised version.

Page 7 line 187: Should this be "extending Eq. (6) to Eq. (7)" ?
Corrected.

Page 7 line 194: replace "site" typo with "in-situ".
Done.

Page 7 line 196: check equation number.
Corrected.

Page 7 Line 197: add (SZA) to DSCD_offset.
Done.

Page 7 line 204: check the Eq. number.
Corrected.

Page 7 line 205: replace "geometry" with "geometric" and cite Brinksma et al, 2008 https://doi.org/10.1029/2007JD008808 and Wagner et al. 2010 https://doi.org/10.5194/amt-3-129-2010
Done. We have added the citation in line 256 of the revised version.

Page 8 line 209-210: the meaning of this sentence is unclear.
We have revised the sentence.

Page 8 line 211: NO2 needs subscript.
Corrected.

Page 8 line 215: missing space after "September 23".
Corrected.

Page 8 line 216: missing period at the end of this sentence.
Corrected.

Page 9 line 233: hyphen needed between "three" and "dimensional"
Done.

Page 9 line 251: delete the extra period.
Done.

Page 10 line 265: change "area" to "areas"
Done.

Page 11 line 294: rewrite the sentence to read "The highest values were between. . ."
The sentence has been rewritten.

Page 11 line 296: see comment for Page 2 line 32-33.
Done.

Page 11 line 300-301: what factors would increase emissions in January compared to October? More home heating?
Central heating from power plant and home heating increase the emissions in January compared to October. We calculated the average $NO_X$ emission rates of four sectors including industry, power, residential, and transportation from the MEIC within the 6th Ring Rd of Beijing in January, September, and October 2012, and the ratio of $NO_X$ emission rates in Jan. and the average in Sep. and Oct (Table S1). The $E_{NOx}$ from power and residential are remarkably higher in January than other two months, especially $E_{NOx}$ from residential in January are about 5 times as much as those in other months. In general, central heating is provided in urban area and the scattered coal combustion for heating is popular in suburb or rural area. Corresponding descriptions are added in lines 371-372 and lines 437-445 of the revised version.

Page 11 line 301-302: were lower PBL heights and smaller wind-speeds found in January compared to September/October?
PBL heights and wind speeds in January are not always smaller than September/October. We have deleted the related description in line 374 of the revised manuscript.

Page 12 line 337: subscript needed on NO2 VCD.

Corrected.

Page 12 line 349: see comment for Page 2 line 32-33.
Done.

Page 13 line 371: considering replacing "residents" with "residential"
Done.

Page 13 lines 375-377: in what way does it indicate the applicability? Please explain/ elaborate.
The sentence is not correct and has been deleted in lines 468-470 of the revised manuscript.

Page 22 623: fix error in reference name.
Done.

Pages 28 and 29: Figures 5 and 6: correct the unit notation (x1016) of VCDs in the caption.
Corrected.

Page 31 Figure 8: the caption says "under three types of wind fields" but only south and north winds are shown.
We have replaced "three types" with "two types".

[revised manuscript text omitted]
 and distribution of four large point $NO_X$ emission sources within  the 6th Ring Rd of Beijing.

840

845

850

855

[Figure]

**Fig. 2** Examples of the NO$_2$ retrieval from two successive spectra measured (**a**) at a 30° elevation angle

(with NO$_2$ differential slant column density (DSCD) of $1.23 \times 10^{17}$ molecules cm$^{-2}$) and (**b**) at a 90° elevation angle (with NO$_2$ DSCD of $6.22 \times 10^{16}$ molecules cm$^{-2}$) on January 18, 2014, at around 11:40 BJT.

[Figure]

**Fig. 3** Time series of the $NO_2$ (**a**) $DSCD_{means}$ (red dots) and (**b**) $DSCD_{offset}$ (black dots) (units of $10^{16}$ molecules $cm^{-2}$) for the 30° elevation angle of each sequence on January 18, 2014. The black curve represents a second-order polynomial fit from individual $DSCD_{offset}$ data points.

890

895

[Figure]

**Fig. 4** Time series of the tropospheric $NO_2$ vertical column density (VCD) for 19 circling journeys on

the Sixth Ring Road of Beijing in January, September, and October, 2014. Lower (upper) error bars and

yellow boxes are the 10th (90th) and 25th (75th) percentiles of the data of each journey, respectively.

Hyphens inside the boxes are the medians, and red circles are the mean values. The numbers of each

900    journey are labeled at the top axis. See Table 1 for detailed information about each journey.

905

910

915

920

[Figure]

**Fig. 5** Distributions of the monthly average $NO_2$ VCD ($10^{16}$ molecules $cm^{-2}$) on the 6th Ring Rd of

945    Beijing in **(a)** January, **(b)** September, and **(c)** October, 2014.

950

955

960

[Figure]

**Fig. 6** Distributions of the maximum and minimum NO$_2$ VCD ($10^{16}$ molecules cm$^{-2}$) on the 6th

Ring Rd of Beijing on the morning of **(a)** January 23 and **(b)** October 13, 2014.

1005

[Figure]

**Fig. 7** Wind fields in Beijing and the surrounding area from ECWRF at 08:00 (left column) and 14:00 (right column) BJT on January 23 and October 13, 2014.

1010

1015

1020

[Figure]

**Fig. 8** Average wind stream and NO$_2$ VCD (E16 molecules cm$^{-2}$) distributions under the  two different types of wind field over Beijing: **(a)** south wind, **(b)** NO$_2$ VCD under south wind, **(c)** north wind, and **(d)** NO$_2$ VCD under north wind.

[Figure]

**Fig. 9** Journey-to-journey variation in **(a)** $F_{NO_2}$ and $E_{NO_x}$, **(b)** $c_\tau$ and $c_L$, **(c)** NO$_2$ VCD and mean wind

speed for  10 circling journeys on the 6th Ring Rd of Beijing in January, September, and October,

1110    2014.

[Figure]

**Fig. 10** Spatial distributions of $NO_X$ emissions (mole $km^{-2}\ h^{-1}$) over Beijing based on the MEIC inventory

in **(a)** January, **(b)** September, and **(c)** October 2012.

1130

1135

1140

[Figure]

**Fig. 11** Journey-to-journey variation in estimated $E_{NO_x}$ and corresponding monthly emissions rates from the MEIC inventory (MEIC_Month) within the 6th Ring Rd of Beijing in January, September, and October 2014. Error bars represent the uncertainties in estimated $E_{NO_x}$

1155

1160

1165

[Figure]

**Fig.S1**  Triple-nested domains of **(a)** the LAPS-WRF-CMAQ model system and **(b)** the distribution of meteorological/environmental monitoring stations.

[Figure]

**Fig.S2**  Scatterplot of simulated wind speed and observations at four stations in Beijing. The standard deviation of the slope is 0.002.

[Figure]

**Fig.S3Fig.3S** Wind rose of simulated wind direction and observations from MICAPS datasets at four stations in Beijing.

[Figure]

**Fig. S4** Time serial of simulated wind speed and direction, and observations during car MAX-DOAS experiments at four weather stations in Beijing.

[Figure]

**Fig.S5** Time series of regional average simulation and in situ observation of $NO_2$ concentration at 12 stations in Beijing.

[Figure]

**Fig.S6** Scatter plot between regional average simulation and observation of $NO_2$ concentration at 12 stations in Beijing.

[Figure]

**Fig. S7** Distributions of the ratio of NOx and NO$_2$ on the 6th Ring Rd of Beijing during five journeys.

[Figure]

**Fig. S8** Same to figure S7, except for the lifetime of NOx (h).

Table S1. Four types of monthly $E_{NOX}$ from the MEIC inventory within the 6th Ring Rd of Beijing in January, September, and October 2012, and the ratio of $E_{NOX}$ in Jan. to the average in Sep. and Oct.

|  | industry | power | resident | transport | total |
|---|---|---|---|---|---|
| January | 5.78 | 1.92 | 1.39 | 3.94 | 13.02 |
| September | 4.06 | 1.15 | 0.25 | 3.93 | 9.40 |
| October | 4.03 | 0.93 | 0.26 | 3.93 | 9.15 |
| Ratio | 1.43 | 1.84 | 5.43 | 1.00 | 1.40 |

---

## Author Comment (AC2) · 17 Aug 2020

**Response to Interactive comments from Anonymous Referee #1**

Referee comments are in black. Author responses are in blue.

General Remarks: The manuscript present the results of observing NO2 emission measurements in Beijing based on the Car MAX-DOAS technology. Through 19 times of city-circle-around Car-MAX-DOAS experiments, the author showed the potential of Car MAX-DOAS measurement technology in atmospheric monitoring. This observation method can be effectively used for dynamic monitoring of urban NO2 emissions. However, the database the authors use for the conclusions is relatively weak. So some revisions are needed to consider this manuscript for publication in ACP.

We thank the anonymous referee for his/her insightful and constructive comments. Below are our point-to-point responses in detail.

Major concerns:

1. Line 24-25, "typically larger NO2 VCD at the southern parts of the 6th Ring Road than at the northern parts". According to Figures 5 and 6, the NO2 VCD at the southern parts and northern parts were not typically different in January.

The sentence has been deleted in the revised version of the manuscript.

2. Since each measurement time was different, from 2 to 2.5 hours (sometimes nearly 3 hours), the author should introduce the traffic situation during the measurement and analyze the impact on the measurement results.

Traffic jam did not occur during the measurement and the impact of traffic situation should be negligible. Different people might drive at different speeds although we had suggested them to drive at a low speed and stable level. Only after the experiments, we realized that driving slowly might also cause a problem for the emission estimate since the change in wind field might be more pronounced with a longer experimental time.

3. The author should introduce the NOx emission sources in Figure 1 and analyze the influence on the measurement.

We have added the related descriptions in lines 198-199 and analyze the influence on the NO2 VCD measurements in lines 396-398 of the revised manuscript.

4. The results of Car Max- DOAS measurement show that NOx emission in heating season is nearly three times as much as that in non heating season, which is obviously higher than that calculated by MEIC inventory estimate. Since central heating is adopted in Beijing urban area, the author should analyze heating season NOx sources in detail, and evaluate the contributions to the measurement results.

Many thanks for suggestions. Central heating from power plant and home heating increase emissions in heating season compared to non-heating season. We calculated the average NO$_X$ emission rates of four sectors including industry, power, residential, and transportation from the MEIC within the 6th Ring Rd of Beijing in January,

September, and October 2012, and the ratio of $NO_X$ emission rates in Jan. and the average in Sep. and Oct (Table S1). The $E_{NOX}$ from power and residential in January are remarkably higher than other two months, especially $E_{NOX}$ from residential in January are about 5 times those in other months. Corresponding descriptions are added in 371-372 and lines 437-445 of the revised version. We cannot retrieve the specific positions of heating season NOx sources from MAX-DOAS measurements by the method used in this study. But we agree that it is a significative scientific issue and will use the source apportionment model to investigate it in the future study.

Minor comments:
1. Abstract, here it is more appropriate to use "different months" instead of "different seasons".
   Corrected.

2. Line 43, "less than" instead of "smaller than"
   Corrected.

3. Line 54-56, "NO and NO2 (together denoted as NOX) form primarily in combustion processes, and the conversion between NO and NO2 in the atmosphere is very rapid" is well known and meaningless here
The sentence has been changed to "the studies on the spatiotemporal variation of NO and NO2 (together denoted as NOX), with the latter being a precursor of nitrate aerosols, are very important for understanding the aerosol formation and its influencing factors" in lines 59-61 of the revised manuscript.

4. Please unify the format of "Car-MAX-DOAS" in the manuscript. For example: line 90 "car-MAX-DOAS", line 94 "Car-MAX-DOAS", line 101 "car MAX-DOAS", line 190 "car MAX-DOAS" et al.
Done.

5. Line 147, "the roof of a car" instead of "the roof a car".
Corrected.

[revised manuscript text omitted]

**Fig.S3Fig.3S** Wind rose of simulated wind direction and observations from MICAPS datasets at four

stations in Beijing.

[Figure]

**Fig. S4** Time serial of simulated wind speed and direction, and observations during car MAX-DOAS experiments at four weather stations in Beijing.

[Figure]

**Fig.S5** Time series of regional average simulation and in situ observation of $NO_2$ concentration at 12 stations in Beijing.

[Figure]

**Fig.S6** Scatter plot between regional average simulation and observation of $NO_2$ concentration at 12 stations in Beijing.

[Figure]

**Fig. S7** Distributions of the ratio of NOx and NO₂ on the 6th Ring Rd of Beijing during five journeys.

[Figure]

**Fig. S8** Same to figure S7, except for the lifetime of NOx (h).

Table S1. Four types of monthly $E_{NOX}$ from the MEIC inventory within the 6th Ring Rd of Beijing in January, September, and October 2012, and the ratio of $E_{NOX}$ in Jan. to the average in Sep. and Oct.

|           | industry | power | resident | transport | total |
|-----------|----------|-------|----------|-----------|-------|
| January   | 5.78     | 1.92  | 1.39     | 3.94      | 13.02 |
| September | 4.06     | 1.15  | 0.25     | 3.93      | 9.40  |
| October   | 4.03     | 0.93  | 0.26     | 3.93      | 9.15  |
| Ratio     | 1.43     | 1.84  | 5.43     | 1.00      | 1.40  |

---

## Author Response (AR2)

**Response to the editor**

**Editor comment:** *The paper is acceptable with a small technical change. At least in the abstract (and perhaps elsewhere), the number of significant figures should be reduced in the emission estimates. For example...given the uncertainty on an individual estimate, the following has excessive significant figures:*

*$22.59 \times 10^{25}$ to $31.28 \times 10^{25}$ molecules s–1,*

*which should be expressed as*

*$22.6 \times 10^{25}$ to $31.3 \times 10^{25}$ molecules s–1,*

*OR*

*$23 \times 10^{25}$ to $31 \times 10^{25}$ molecules s–1,*

*These numbers should be changed throughout the abstract and perhaps in the paper as well.*

**Author Response:** We thank the editor, Prof. Robert McLaren, for handling this manuscript and giving us constructive suggestions. We have revised all the numbers of significant figures throughout the paper accordingly. Typos have also been corrected as marked up in the revision-tracking version of the manuscript below.

[revised manuscript text omitted]

MAX-DOAS experiments at four weather stations in Beijing.

[Figure]

**Figure S5.**  Time series of regional average simulation and in situ observation of $NO_2$ concentration at 12 stations in Beijing.

[Figure]

**Figure S6.**  Scatter plot between regional average simulation and observation of $NO_2$ concentration at 12 stations in Beijing.

[Figure]

**Figure S7. ** Distributions of the ratio of NOx and NO₂ on the 6th Ring Rd of Beijing during five journeys.

[Figure]

**Figure S8.**  Same to figure S7, except for the lifetime of NOx (h).

**Table S1.** Four types of monthly $E_{NOX}$ from the MEIC inventory within the 6th Ring Rd of Beijing in January, September, and October 2012, and the ratio of $E_{NOX}$ in Jan. to the average in Sep. and Oct.

|           | industry | power | resident | transport | total |
|-----------|----------|-------|----------|-----------|-------|
| January   | 5.78     | 1.92  | 1.39     | 3.94      | 13.02 |
| September | 4.06     | 1.15  | 0.25     | 3.93      | 9.40  |
| October   | 4.03     | 0.93  | 0.26     | 3.93      | 9.15  |
| Ratio     | 1.43     | 1.84  | 5.43     | 1.00      | 1.40  |